# Dynamic Continuous Hyperparameter Tuning for Generalized Linear Contextual Bandits

## Abstract

In stochastic contextual bandits, an agent sequentially makes actions from a time-dependent action set based on past experience to minimize the cumulative regret. Like many other machine learning algorithms, the performance of bandits heavily depends on the values of hyperparameters, and theoretically derived parameter values may lead to unsatisfactory results in practice. Moreover, it is infeasible to use offline tuning methods like cross-validation to choose hyperparameters under the bandit environment, as the decisions should be made in real time. To address this challenge, we propose the first online continuous hyperparameter tuning framework for contextual bandits to learn the optimal parameter configuration within a search space on the fly. Specifically, we use a double-layer bandit framework named CDT (Continuous Dynamic Tuning) and formulate the hyperparameter optimization as a non-stationary continuum-armed bandit, where each arm represents a combination of hyperparameters, and the corresponding reward is the algorithmic result. For the top layer, we propose the Zooming TS algorithm that utilizes Thompson Sampling (TS) for exploration and a restart technique to get around the *switching* environment. The proposed CDT framework can be easily utilized to tune contextual bandit algorithms without any pre-specified candidate set for multiple hyperparameters. We further show that it could achieve a sublinear regret in theory and performs consistently better than all existing methods on both synthetic and real datasets.

## 1 Introduction

The contextual bandit is a powerful framework for modeling sequential learning problems under uncertainty, with substantial applications in recommendation systems Li et al. (2010), clinical trials Woodroofe (1979), personalized medicine Bastani & Bayati (2020), etc. At each round $t$, the agent sequentially interacts with the environment by pulling an arm from a feasible arm set $\mathcal{A}_t$ of $K$ arms ($K$ might be infinite), where every arm could be represented by a $d$-dimensional feature vector, and only the reward of the selected arm is revealed. Here $\mathcal{A}_t$ is drawn IID from an unknown distribution. In order to maximize the cumulative reward, the agent would update its strategy on the fly to balance the exploration-exploitation tradeoff.

Generalized linear bandit (GLB) was first proposed in Filippi et al. (2010) and has been extensively studied under various settings over the recent years Jun et al. (2017); Kang et al. (2022), where the stochastic payoff of an arm follows a generalized linear model (GLM) of its associated feature vector and some fixed, but initially unknown parameter $\theta^*$. Note that GLB extends the linear bandit Abbasi-Yadkori et al. (2011) in representation power and has greater applicability in the real world, e.g. logistic bandit algorithms can achieve improvement over linear bandit when the rewards are binary. Upper Confidence Bound (UCB) Auer et al. (2002a); Filippi et al. (2010); Li et al. (2010) and Thompson Sampling (TS) Agrawal & Goyal (2012; 2013) are the two most popular ideas to solve the GLB problem. Both of these methods could achieve the optimal regret bound of order $\tilde{O}(\sqrt{T})$[1] under some mild conditions, where $T$ stands for the total number of rounds Agrawal & Goyal (2013).

However, the empirical performance of these bandit algorithms significantly depends on the configuration of hyperparameters, and simply using theoretical optimal values often yields unsatisfactory practical results, not to mention some of them are unspecified and needed to be learned in reality. For

---

[1] $\tilde{O}(\cdot)$ ignores the poly-logarithmic factors.

example, in both LinUCB Li et al. (2010) and LinTS Abeille & Lazaric (2017); Agrawal & Goyal (2013) algorithms, there are hyperparameters called exploration rates that govern the tradeoff and hence the learning process. But it has been verified that the best exploration rate to use is always instance-dependent and may vary at different iterations Bouneffouf & Claeys (2020); Ding et al. (2022b). Note it is inherently impossible to use any state-of-the-art offline hyperparameter tuning methods such as cross validation Stone (1974) or Bayesian optimization Frazier (2018) since decisions in bandits should be made in real time. To choose the best hyperparameters, some previous works use grid search in their experiments Ding et al. (2021); Jun et al. (2019), but obviously, this approach is infeasible when it comes to reality, and how to manually discretize the hyperparameter space is also unclear. Conclusively, this limitation has already become a bottleneck for bandit algorithms in real-world applications, but unfortunately, it has rarely been studied in the previous literature.

The problem of hyperparameter optimization for contextual bandits was first studied in Bouneffouf & Claeys (2020), where the authors proposed two methods named OPLINUCB and DOPLINUCB to learn the optimal exploration rate of LinUCB in a finite candidate set by viewing each candidate as an arm and then using multi-armed bandit to pull the best one. However, 1) the authors did not provide any theoretical support for these methods, and 2) we believe the best exploration parameter would vary during the iterations – more exploration may be preferred at the beginning due to the lack of observations, while more exploitation would be favorable in the long run when the model estimate becomes more accurate. Furthermore, 3) they only consider tuning one single hyperparameter. To tackle these issues, Ding et al. (2022b) proposed TL and Syndicated framework by using a non-stationary multi-armed bandit for the hyperparameter set. However, their approach still requires a pre-defined set of hyperparameter candidates. In practice, choosing the candidates requires domain knowledge and plays a crucial role in the performance. Also, using a piecewise-stationary setting instead of a complete adversarial bandit (e.g. EXP3) for hyperparameter tuning is more efficient since we expect a fixed hyperparameter setting would yield indistinguishable results in a period of time. Conclusively, it would be more efficient to use a continuous space for bandit hyperparameter tuning.

We propose an efficient bandit-over-bandit (BOB) framework Cheung et al. (2019) named Continuous Dynamic Tuning (CDT) framework for bandit hyperparameter tuning in the continuous hyperparameter space, without requiring a pre-defined set of hyperparameter configurations. For the top layer bandit we formulate the online hyperparameter tuning as a non-stationary Lipschitz continuum-arm bandit problem with noise where each arm represents a hyperparameter configuration and the corresponding reward is the performance of the GLB, and the expected reward is a time-dependent Lipschitz function of the arm with some biased noise. Here the bias depends on the previous observations since the history could also affect the update of bandit algorithms. It is also reasonable to assume the Lipschitz functions are piecewise stationary since we believe the expected reward would be stationary with the same hyperparameter configuration over a period of time (i.e. *switching* environment). Specifically, for the top layer of our CDT framework, we propose the Zooming TS algorithm with Restarts, and the key idea is to adaptively refine the hyperparameter space and zoom into the regions with more promising reward Kleinberg et al. (2019) by using the TS methodology Chapelle & Li (2011). Moreover, the restarts could handle the piecewise changes of the bandit environments. We summarize our contributions as follows:

1) We propose an online continuous hyperparameter optimization framework for contextual bandits called CDT that handles all aforementioned issues of previous methods with theoretical guarantees. To the best of our knowledge, CDT is the first hyperparameter tuning method (even model selection method) with continuous candidates in the bandit community. 2) For the top layer of CDT, we propose the Zooming TS algorithm with Restarts for Lipschitz bandits under the *switching* environment. To the best of our knowledge, our work is the first one to consider the Lipschitz bandits under the *switching* environment, and the first one to utilize TS methodology in Lipschitz bandits. 3) Experiments on both synthetic and real datasets with various GLBs validate the efficiency of our method.

**Notations:** For a vector $x \in \mathbb{R}^d$, we use $\|x\|$ to denote its $l_2$ norm and $\|x\|_A := \sqrt{x^T A x}$ for any positive definite matrix $A \in \mathbb{R}^{d \times d}$. We also denote $[T] = \{1, \dots, T\}$ for $T \in \mathbb{N}^+$.

## 2 RELATED WORK

There has been extensive literature on contextual bandit algorithms, and most of them are based on the UCB or TS techniques. For example, several UCB-type algorithms have been proposed for GLB, such

as GLM-UCB Filippi et al. (2010) and UCB-GLM Li et al. (2017) that achieve the optimal $\tilde{O}(\sqrt{T})$ regret bound. Another rich line of work on GLBs follows the TS idea, including Laplace-TS Chapelle & Li (2011), SGD-TS Ding et al. (2021), etc. In this paper, we focus on the hyperparameter tuning of contextual bandits, which is a practical but under-explored problem. For related work, Sharaf & Daumé III (2019) first studied how to learn the exploration parameters in contextual bandits via a meta-learning method. However, this algorithm fails to adjust the learning process based on previous observations and hence can be unstable in practice. Bouneffouf & Claeys (2020) then proposed OPLINUCB and DOPLINUCB to choose the exploration rate of LinUCB from a candidate set, and moreover Ding et al. (2022b) formulates the hyperparameter tuning problem as a non-stochastic multi-armed bandit and utilizes the classic EXP3 algorithm. However, as we mentioned in Section 1, both works have several limitations that could be improved. Note that hyperparameter tuning could be regarded as a branch of model selection in bandit algorithms. For this general problem, Agarwal et al. (2017) proposed a master algorithm that could combine multiple bandit algorithms, while Foster et al. (2019) initiated the study of model selection tradeoff in contextual bandits and proposed the first model selection algorithm for contextual linear bandits. We further explain why these general model selection methods fail for the bandit hyperparameter tuning task in Remark B.1 in Appendix B due to the space limit. In contrast, we propose the first continuous hyperparameter tuning framework for contextual bandits, which doesn't require a pre-defined set of candidates.

We also briefly review the literature on Lipschitz bandits that follows two key ideas. One is uniformly discretizing the action space into a mesh Kleinberg (2004); Magureanu et al. (2014) so that any learning process like UCB could be directly utilized. Another more popular idea is adaptive discretization on the action space by placing more probes in more encouraging regions Bubeck et al. (2008); Kleinberg et al. (2019); Lu et al. (2019); Valko et al. (2013), and UCB could be used for exploration. Furthermore, the Lipschitz bandit under adversarial corruptions was recently studied in Kang et al. (2023). In addition, Podimata & Slivkins (2021) proposed the first fully adversarial Lipschitz bandit in an adaptive refinement manner and derived instance-dependent regret bounds, but their algorithm relies on some unspecified hyperparameters and is computationally infeasible. Since the expected reward function for hyperparameters would not drastically change every time, it is also inefficient to use a fully adversarial algorithm here. Therefore, we introduce a new problem of Lipschitz bandits under the *switching* environment, and propose the Zooming TS algorithm with a restart trick to deal with the "almost stationary" nature of the bandit hyperparameter tuning problem.

## 3  PRELIMINARIES

We first review the problem setting of contextual bandit algorithms. Denote $T$ as the total number of rounds and $K$ as the number of arms we could choose at each round, where $K$ could be infinite. At each round $t \in [T] := \{1, \ldots, T\}$, the player is given $K$ arms represented by a set of feature vectors $\mathcal{X}_t = \{x_{t,a} \mid a \in [K]\} \in \mathbb{R}^d$, where $x_{t,a}$ is a $d$-dimensional vector containing information of arm $a$ at round $t$. The player selects an action $a_t \in [K]$ based on the current $\mathcal{X}_t$ and previous observations, and only receives the payoff of the pulled arm $a_t$. Denote $x_t := x_{t,a_t}$ as the feature vector of the chosen arm $a_t$ and $y_t$ as the corresponding reward. We assume the reward $y_t$ follows a canonical exponential family with minimal representation, a.k.a. generalized linear bandits (GLB) with some mean function $\mu(\cdot)$. In addition, one can represent this model by $y_t = \mu(x_t^\top \theta^*) + \epsilon_t$, where $\epsilon_t$ follows a sub-Gaussian distribution with parameter $\sigma^2$ independent with the previous information filtration $\mathcal{F}_t = \sigma(\{a_s, x_s, y_s\}_{s=1}^{t-1})$ and the sigma field $\sigma(\{x_t\})$, and $\theta^*$ is some unknown coefficient. Denote $a_{t,*} := \arg\max_{a \in [K]} \mu(x_{t,a}^\top \theta^*)$ as the optimal arm at round $t$ and $x_{t,*}$ as its corresponding feature vector. The goal is to minimize the expected cumulative regret defined as:

$$R(T) = \sum_{t=1}^{T} \left[ \mu(x_{t,*}^\top \theta^*) - \mathbb{E}\left( \mu(x_t^\top \theta^*) \right) \right]. \tag{1}$$

Note that all state-of-the-art contextual GLB algorithms depend on at least one hyperparameter to balance the well-known exploration-exploitation tradeoff. For example, LinUCB Li et al. (2010), the most popular UCB linear bandit, uses the following rule for arm selection at round $t$:

$$a_t = \arg\max_{a \in [K]} x_{t,a}^\top \hat{\theta}_t + \alpha_1(t) \left\| x_{t,a} \right\|_{V_t^{-1}}. \tag{LinUCB}$$

Here the model parameter $\hat{\theta}_t$ is estimated at each round $t$ via ridge regression, i.e. $\hat{\theta}_t = V_t^{-1} \sum_{s=1}^{t-1} x_s y_s$ where $V_t = \lambda I_r + \sum_{s=1}^{t-1} x_s x_s^\top$. And it considers the standard deviation of each

arm with an exploration parameter $\alpha_1(t)$, where with a larger value of $\alpha_1(t)$ the algorithm will be more likely to explore uncertain arms. Note that the regularization parameter $\lambda$ is only used to ensure $V_t$ is invertible and hence its value is not crucial and commonly set to 1. In theory we can choose the value of $\alpha_1(t)$ as $\alpha_1(t) = \sigma\sqrt{r\log\left((1+t/\lambda)/\delta\right)} + \|\theta^*\|\sqrt{\lambda}$, to achieve the optimal $\widetilde{O}(\sqrt{T})$ bound of regret: However, in practice, the values of $\sigma$ and $\|\theta^*\|$ are unspecified, and hence this theoretical value of $\alpha_1(t)$ is inaccessible. Furthermore, it has been shown that this is a very conservative choice that would lead to unsatisfactory practical performance, and the optimal hyperparameter values to use are distinct and far from the theoretical ones under different algorithms or settings. We also conduct a series of simulations with several state-of-the-art GLB algorithms to validate this fact, which is deferred to Appendix A.1. Conclusively, the best exploration parameter to use in practice should always be chosen dynamically based on the specific scenario and past observations. In addition, many GLB algorithms depend on some other hyperparameters, which may also affect the performance. For example, SGD-TS also involves a stepsize parameter for the stochastic gradient descent besides the exploration rate, and it is well known that a decent stepsize could remarkably accelerate the convergence Loizou et al. (2021). To handle all these cases, we propose a general framework that can be used to automatically tune multiple continuous hyperparameters for a contextual bandit.

For a certain contextual bandit, assume there are $p$ different hyperparameters $\alpha(t) = \{\alpha_i(t)\}_{i=1}^p$, and each hyperparameter $\alpha_i(t)$ could take values in an interval $[a_i, b_i]$, $\forall t$. Denote the parameter space $A = \bigotimes_{i=1}^p [a_i, b_i]$, and the theoretical optimal values as $\alpha^*(t)$. Given the past observations $\mathcal{F}_t$ by round $t$, we write $a_t(\alpha(t)|\mathcal{F}_t)$ as the arm we pulled when the hyperparameters are set to $\alpha(t)$, and $x_t(\alpha(t)|\mathcal{F}_t)$ as the corresponding feature vector.

The main idea of our algorithm is to formulate the hyperparameter optimization as a (another layer of) non-stationary Lipschitz bandit in the continuous space $A \subseteq \mathbb{R}^p$, i.e. the agent chooses an arm (hyperparameter combination) $\alpha \in A$ in round $t \in [T]$, and then we decompose $\mu(x_t(\alpha|\mathcal{F}_t)^\top \theta^*)$ as

$$\mu(x_t(\alpha|\mathcal{F}_t)^\top \theta^*) = g_t(\alpha) + \eta_{\mathcal{F}_t,\alpha}. \tag{2}$$

Here $g_t$ is some time-dependent Lipschitz function that formulates the performance of the bandit algorithm under the hyperparameter combination $\alpha$ at round $t$, since the bandit algorithm tends to pull similar arms if the chosen values of hyperparameters are close at round $t$. To demonstrate that our Lipschitz assumption w.r.t. the hyperparameter values in Eqn. (3) is reasonable, we conduct simulations with LinUCB and LinTS, and defer it to Appendix A due to the space limit. Moreover, $(\eta_{\mathcal{F}_t,\alpha} - \mathbb{E}[\eta_{\mathcal{F}_t,\alpha}])$ is IID sub-Gaussian with parameter $\tau^2$, and to be fair we assume $\mathbb{E}[\eta_{\mathcal{F}_t,\alpha}]$ could also depend on the history $\mathcal{F}_t$ since past observations would explicitly influence the model parameter estimation and hence the decision making at each round. In addition to Lipschitzness, we also suppose $g_t$ follows a *switching* environment: $g_t$ is piecewise stationary with some change points, i.e.

$$|g_t(\alpha_1) - g_t(\alpha_2)| \leq \|\alpha_1 - \alpha_2\|, \; \forall \alpha_1, \alpha_2 \in A; \tag{3}$$

$$\sum_{t=1}^{T-1} \mathbf{1}[\exists \alpha \in A : g_t(\alpha) \neq g_{t+1}(\alpha)] = c(T), c(T) \in \mathbb{N}. \tag{4}$$

Since after sufficient exploration, the expected reward should be stable with the same hyperparameter setting, we could assume that $c(T) = \tilde{O}(1)$. More justification on this piecewise Lipschitz assumption is deferred to Remark B.2 in Appendix B due to the limited space. Although numerous research works have considered the *switching* environment (a.k.a. *abruptly-changing* environment) for multi-armed or linear bandits Auer et al. (2002b); Wei et al. (2016), our work is the first to introduce this setting into the continuum-armed bandits. In Section 4.1, we will show that by combining our proposed Zooming TS algorithm for Lipschitz bandits with a simple restarted strategy, a decent regret bound could be achieved under the *switching* environment.

## 4 MAIN RESULTS

In this section, we present our novel online hyperparameter optimization framework that could be easily adapted to most contextual bandit algorithms. We first introduce the continuum-arm Lipschitz bandit problem under the *switching* environment, and propose the Zooming TS algorithm with Restarts which modifies the traditional Zooming algorithm Kleinberg et al. (2019) to make it more efficient and also adaptive to the *switching* environment. Subsequently, we propose our bandit

---

**Algorithm 1** Zooming TS algorithm with Restarts

---

**Input:** Time horizon $T$, space $A$, epoch size $H$.

1: **for** $t = 1$ **to** $T$ **do**
2:     **if** $t \in \{\tau H + 1 : \tau = 0, 1, \dots \}$ **then**
3:         Initialize the total candidate space $A_0 = A$ and the active set $J \subseteq A_0$ s.t. $A_0 \subseteq \cup_{v \in J} B(v, r_1(v))$ and $n_1(v) \leftarrow 1, \forall v \in J$.     ▷Restart
4:     **else if** $\hat{f}_t(v) - \hat{f}_t(u) > r_t(v) + 2r_t(u)$ for some pair of $u, v \in J$ **then**
5:         Set $J = J \backslash \{u\}$ and $A_0 = A_0 \backslash B(u, r_t(u))$.     ▷Removal
6:     **end if**
7:     **if** $A_0 \not\subseteq \cup_{v \in J} B(v, r_t(v))$ **then**     ▷Activation
8:         Activate and pull some point $v \in A_0$ that has not been covered: $J = J \cup \{v\}, v_t = v$.
9:     **else**
10:         $v_t = \operatorname{argmax}_{v \in J} I_t(v)$, break ties arbitrarily.     ▷Selection
11:     **end if**
12:     Observe the reward $\tilde{y}_{t+1}$, and then update components in the Zooming TS algorithm: $n_{t+1}(v), \hat{f}_{t+1}(v), r_{t+1}(v), s_{t+1}(v)$ for the chosen $v_t \in J$:

$$n_{t+1}(v_t) = n_t(v_t) + 1, \hat{f}_{t+1}(v_t) = (\hat{f}_t(v_t)n_t(v_t) + \tilde{y}_{t+1})/n_{t+1}(v_t).$$

13: **end for**

---

hyperparameter tuning framework named Continuous Dynamic Tuning (CDT) by making use of our proposed Zooming TS algorithm with Restarts and the Bandit-over-Bandit (BOB) idea.

W.l.o.g we assume that there exists a positive constant $S$ such that $\|\theta^*\| \leq S$ and $\|x_{t,a}\| \leq 1$, $\forall t, a$, and each hyperparameter space has been shifted and scaled to $[0, 1]$. We also assume that the mean reward $\mu(x_{t,a}^\top \theta^*) \in [0, 1]$, and hence naturally $g_t(\alpha) \in [0, 1]$, $\forall \alpha \in A = [0, 1]^p, t \in [T]$.

## 4.1 ZOOMING TS ALGORITHM WITH RESTARTS

For simplicity, we will reload some notations in this subsection. Consider the non-stationary Lipschitz bandit problem on a compact space $A$ under some metric $\mathrm{Dist}(\cdot, \cdot) \geq 0$, where the covering dimension is denoted by $p_c$. The learner pulls an arm $v_t \in A$ at round $t \in [T]$ and subsequently receives a reward $\tilde{y}_t$ sampled independently of $\mathbb{P}_{v_t}$ as $\tilde{y}_t = f_t(v_t) + \eta_v$, where $t = 1, \dots, T$ and $\eta_v$ is IID zero-mean error with sub-Guassian parameter $\tau_0^2$, and $f_t$ is the expected reward function at round $t$ and is Lipschitz with respect to $\mathrm{Dist}(\cdot, \cdot)$. The *switching* environment assumes the time horizon $T$ is partitioned into $c(T) + 1$ intervals, and the bandit stays stationary within each interval, i.e.

$$|f_t(m) - f_t(n)| \leq \mathrm{Dist}(m, n), \quad m, n \in A; \text{ and } \sum_{t=1}^{T-1} \mathbf{1}[\exists m \in A : f_t(m) \neq f_{t+1}(m)] = c(T).$$

Here in this section $c(T) = o(T)$ could be any integer. The goal of the learner is to minimize the expected (dynamic) regret that is defined as:

$$R_L(T) = \sum_{t=1}^{T} \max_{v \in A} f_t(v) - \sum_{t=1}^{T} \mathbb{E}\left(f_t(v_t)\right).$$

At each round $t$, $v_t^* := \operatorname{argmax}_{v \in A} f_t(v)$ denotes the maximal point (w.l.o.g. assume it's unique), and $\Delta_t(v) = f_t(v^*) - f_t(v)$ is the "badness" of each arm $v$. We also denote $A_{r,t}$ as the $r$-optimal region at the scale $r \in (0, 1]$, i.e. $A_{r,t} = \{v \in A : r/2 < \Delta_t(v) \leq r\}$ at time $t$. Then the $r$-zooming number $N_{z,t}(r)$ of $(A, f_t)$ is defined as the minimal number of balls of radius no more than $r$ required to cover $A_{r,t}$. (Note the subscript $z$ stands for zooming here.) Next, we define the zooming dimension $p_{z,t}$ Kleinberg et al. (2019) at time $t$ as the smallest $q \geq 0$ such that for every $r \in (0, 1]$ the $r$-zooming number can be upper bounded by $cr^{-q}$ for some multiplier $c > 0$ free of $r$:

$$p_{z,t} = \min\{q \geq 0 : \exists c > 0, N_{z,t}(r) \leq cr^{-q}, \forall r \in (0, 1]\}.$$

It's obvious that $0 \leq p_{z,t} \leq p_c$, $\forall t \in [T]$. (Note $p_{z,t}$ is fixed under the stationary environment.) On the other hand, the zooming dimension could be much smaller than $p_c$ under some mild conditions. For

example, if the payoff function $f_t$ defined on $\mathbb{R}^{p_c}$ is greater than $\|v_t^* - v\|^\beta$ in scale for some $\beta \geq 1$ around $v^*$ in the space $A$, i.e. $f_t(v_t^*) - f_t(v) = \Omega(\|v_t^* - v\|^\beta)$, then it holds that $p_{z,t} \leq (1 - 1/\beta)p_c$. Note that we have $\beta = 2$ (i.e. $p_{z,t} \leq p_c/2$) when $f_t(\cdot)$ is $C^2$-smooth and strongly concave in a neighborhood of $v^*$. More details are presented in Appendix C. Since the expected reward Lipschitz function $f_t(\cdot)$ is fixed in each time interval under the *switching* environment, the zooming number and zooming dimension $p_{z,t}$ would also stay identical. And we also write $p_{z,*} = \max_{t\in[T]} p_{z,t} \leq p_c$.

Our proposed Algorithm 1 extends the classic Zooming algorithm Kleinberg et al. (2019), which was used under the stationary Lipschitz bandit environment, by adding two new ingredients for better efficiency and adaptivity to non-stationary environment: on the one hand, we employ the TS methodology and propose a novel removal step. Here we utilize TS since it was shown that TS is more robust than UCB in practice Chapelle & Li (2011); Wang & Chen (2018), and the removal procedure could adaptively subtract regions that are prone to yield low rewards. Both of these two ideas could enhance the algorithmic efficiency, which coincides with the practical orientation of our work. On the other hand, the restarted strategy proceeds our proposed Zooming TS in epochs and refreshes the algorithm after every $H$ rounds. The epoch size $H$ is fixed through the total time horizon and controls the tradeoff between non-stationarity and stability. Note that $H$ in our algorithm does not need to match the actual length of stationary intervals of the environment, and we would discuss its selection later. At each epoch, we maintain a time-varying active arm set $S_t \subseteq A$, which is initially empty and updated every time. For each arm $v \in A$ and time $t$, denote $n_t(v)$ as the number of times arm $v$ has been played before time $t$ since the last restart, and $\hat{f}_t(v)$ as the corresponding average sample reward. We let $\hat{f}_t(v) = 0$ when $n_t(v) = 0$. Define the confidence radius and the TS standard deviation of active arm $v$ at time $t$ respectively as

$$r_t(v) = \sqrt{\frac{13\tau_0^2 \ln T}{2n_t(v)}}, \quad s_t(v) = s_0\sqrt{\frac{1}{n_t(v)}}, \tag{5}$$

where $s_0 = \sqrt{52\pi\tau_0^2 \ln(T)}$. We call $B(v, r_t(v)) = \{u \in \mathbb{R}^p : \text{Dist}(u, v) \leq r_t(v)\}$ as the confidence ball of arm $v$ at time $t \in [T]$. We construct a randomized algorithm by choosing the best active arm according to the perturbed estimate mean $I_t(\cdot)$:

$$I_t(v) = \hat{f}_t(v) + s_t(v)Z_{t,v}, \tag{6}$$

where $Z_{t,v}$ is i.i.d. drawn from the clipped standard normal distribution: we first sample $\tilde{Z}_{t,v}$ from the standard normal distribution and then set $Z_{t,v} = \max\{1/\sqrt{2\pi}, \tilde{Z}_{t,v}\}$. This truncation was also used in TS multi-armed bandits Jin et al. (2021), and our algorithm clips the posterior samples with a lower threshold to avoid underestimation of good arms. Moreover, the explanations of the TS update is deferred to Appendix D due to the space limit.

The regret analysis of Algorithm 1 is very challenging since the active arm set is constantly changing and the optimal arm $v^*$ cannot be exactly recovered under the Lipschitz bandit setting. Thus, existing theory on multi-armed bandits with TS is not applicable here. We overcome these difficulties with some innovative use of metric entropy theory, and the regret bound of Algorithm 1 is given as follows.

**Theorem 4.1.** *With $H = \Theta\left((T/c(T))^{(p_{z,*}+2)/(p_{z,*}+3)}\right])$, the total regret of our Zooming TS algorithm with Restarts under the switching environment over time $T$ is bounded as*

$$R_L(T) \leq \tilde{O}\left((c(T))^{1/(p_{z,*}+3)} \, T^{(p_{z,*}+2)/(p_{z,*}+3)}\right),$$

*when $c(T) > 0$. In addition, if the environment is stationary (i.e. $c(T) = 0, f_t = f, p_{z,t} = p_{z,*} := p_z, \forall t \in [T]$), then by using $H = T$ (i.e. no restart), our Zooming TS algorithm could achieve the optimal regret bound for Lipschitz bandits up to logarithmic factors:*

$$R_L(T) \leq \tilde{O}\left(T^{(p_z+1)/(p_z+2)}\right).$$

We also present empirical studies to further evaluate the performance of our Algorithm 1 compared with stochastic Lipschitz bandit algorithms in Appendix A.3. A potential drawback of Theorem 4.1 is that the optimal epoch size $H$ under *switching* environment relies on the value of $c(T)$ and $p_{z,*}$, which are unspecified in reality. However, this problem could be solved by using the BOB idea Cheung et al. (2019); Zhao et al. (2020) to adaptively choose the optimal epoch size with a meta

algorithm (e.g. EXP3 Auer et al. (2002b)) in real time. In this case, we prove the expected regret can be bounded by the order of $\tilde{O}\left(T^{\frac{p_c+2}{p_c+3}} \cdot \max\left\{c(T)^{\frac{1}{p_c+3}}, T^{\frac{1}{(p_c+3)(p_c+4)}}\right\}\right)$ in general, and some better regret bounds in problem-dependent cases. More details are presented in Theorem F.1 with its proof in Appendix F. However, in the following Section 4.2 we could simply set $H = T^{(2+p)/(3+p)}$ in our CDT framework where $p$ is the number of hyperparameters to be tuned after assuming $c(T) = \tilde{O}(1)$ is of constant scale up to logarithmic terms. Note our work introduces a new problem on Lipschitz bandits under the *switching* environment. One potential limitation of our work is that how to deduce a regret lower bound under this problem setting is unclear, and we leave it as a future direction.

## 4.2 ONLINE CONTINUOUS HYPERPARAMETER OPTIMIZATION FOR CONTEXTUAL BANDITS

Based on the proposed algorithm in the previous subsection, we introduce our online double-layer Continuous Dynamic Tuning (CDT) framework for hyperparameter optimization of contextual bandit algorithms. We assume the arm to be pulled follows a fix distribution given the hyperparameters to be used and the history at each round. The detailed algorithm is shown in Algorithm 2. Our method extends the bandit-over-bandit (BOB) idea that was first proposed for non-stationary stochastic bandit problems Cheung et al. (2019), where it adjusts the sliding-window size dynamically based on the changing model. In our work, for the top layer we use our proposed Algorithm 1 to tune the best hyperparameter values from the admissible space, where each arm represents a hyperparameter configuration and the corresponding reward is the algorithmic result. $T_2$ is the length of each epoch (i.e. $H$ in Algorithm 1), and we would refresh our Zooming TS Lipschitz bandit after every $T_2$ rounds as shown in Line 5 of Algorithm 2 due to the non-stationarity. The bottom layer is the primary contextual bandit and would run with the hyperparameter values $\alpha(i_t)$ chosen from the top layer at each round $t$. We also include a warming-up period of length $T_1$ in the beginning to guarantee sufficient exploration as in Li et al. (2017); Ding et al. (2021). Despite the focus of our CDT framework is on the practical aspect, we also present a novel theoretical analysis in the following.

Although there has been a rich line of work on regret analysis of UCB and TS GLB algorithms, most literature certainly requires that some hyperparameters, e.g. exploration rate, always take their theoretical values. It is challenging to study the regret bound of GLB algorithms when their hyperparameters are synchronously tuned in real time, since the chosen hyperparameter values may be far from the theoretical ones in practice, not to mention that previous decisions would also affect the current update cumulatively. Moreover, there is currently no existing literature and regret analysis on hyperparameter tuning (or model selection) for bandit algorithms with an infinite number of candidates in a continuous space. Recall that we denote $\mathcal{F}_t = \sigma(\{a_s, X_s, y_s\}_{s=1}^{t-1})$ as the past information before round $t$ under our CDT framework, and $a_t, x_t$ are the chosen arm and its corresponding feature vector at time $t$, which implies that $a_t = a_t(\alpha(i_t)|\mathcal{F}_t), x_t = x_t(\alpha(i_t)|\mathcal{F}_t)$. Furthermore, we denote $\alpha^*(t)$ as the theoretical optimal value at round $t$ and $\mathcal{F}_t^*$ as the past information filtration by always using the theoretical optimal $\alpha^*(t)$. Since the decision at each round $t$ also depends on the history observe by time $t$, the pulled arm with the same hyperparameter $\alpha(t)$ might be different under $\mathcal{F}_t$ or $\mathcal{F}_t^*$. To analyze the cumulative regret $R(T)$ of our Algorithm 2, we first decompose it into four quantities:

$$R(T) = \underbrace{\mathbb{E}\left[\sum_{t=1}^{T_1}\left(\mu(x_{t,*}^\top\theta^*) - \mu(x_t^\top\theta^*)\right)\right]}_{\text{Quantity (A)}} + \underbrace{\mathbb{E}\left[\sum_{t=T_1+1}^{T}\left(\mu(x_{t,*}^\top\theta^*) - \mu(x_t(\alpha^*(t)|\mathcal{F}_t^*)^\top\theta^*)\right)\right]}_{\text{Quantity (B)}}$$

$$+ \underbrace{\mathbb{E}\left[\sum_{t=T_1+1}^{T}\left(\mu\left(x_t(\alpha^*(t)|\mathcal{F}_t^*)^\top\theta^*\right) - \mu(x_t(\alpha^*(t)|\mathcal{F}_t)^\top\theta^*)\right)\right]}_{\text{Quantity (C)}}$$

$$+ \underbrace{\mathbb{E}\left[\sum_{t=T_1+1}^{T}\left(\mu\left(x_t(\alpha^*(t)|\mathcal{F}_t)^\top\theta^*\right) - \mu(x_t(\alpha(i_t)|\mathcal{F}_t)^\top\theta^*)\right)\right]}_{\text{Quantity (D)}}.$$

Intuitively, Quantity (A) is the regret paid for pure exploration during the warming-up period and could be controlled by the order $O(T_1)$. Quantity (B) is the regret of the contextual bandit algorithm

---

**Algorithm 2** Continuous Dynamic Tuning (CDT)

---

**Input:** $T_1, T_2, \{\mathcal{X}_t\}_{t=1}^{T}, A = \bigotimes_{i=1}^{p}[a_i, b_i]$.

1: Randomly choose $a_t \in [K]$ and observe $x_t, y_t$, $t \leq T_1$.
2: Initialize the hyperparameter active set $J$ s.t. $A \subseteq \cup_{v \in J} B(v, r_1(v))$ where $n_{T_1}(v) \leftarrow 1, \forall v \in J$.
3: **for** $t = (T_1 + 1)$ **to** $T$ **do**
4:     Run the $t$-th iteration of Algorithm 1 with initial input horizon $T - T_1$, input space $A$ and restarting epoch length $T_2$. Denote the pulled arm at round $t$ as $\alpha(i_t) \in A$.    ▷Top
5:     Run the contextual bandit algorithm with hyperparameter $\alpha(i_t)$ to pull an arm $a_t$. ▷Bottom
6:     Obtain $y_t$ and update components in the contextual bandit algorithm.    ▷Bottom Update
7:     Update components in Algorithm 1 by treating $y_t$ as the reward of arm $\alpha(i_t)$  ▷Top Update
8: **end for**

---

that runs with the theoretical optimal hyperparameters $\alpha^*(t)$ all the time, and hence it could be easily bounded by the optimal scale $\tilde{O}(\sqrt{T})$ based on the literature. Quantity (C) is the difference of cumulative reward with the same $\alpha^*(t)$ under two separate lines of history. Quantity (D) is the extra regret paid to tune the hyperparameters on the fly. By using the same line of history $\mathcal{F}_t$ in Quantity (D), the regret of our Zooming TS algorithm with Restarts in Theorem 4.1 can be directly used to bound Quantity (D). Conclusively, we deduce the following theorem:

**Theorem 4.2.** *Under our problem setting in Section 3, for UCB and TS GLB algorithms with exploration hyperparameters (e.g. LinUCB, UCB-GLM, GLM-UCB, LinTS), by taking $T_1 = O(T^{2/(p+3)}), T_2 = O(T^{(p+2)/(p+3)})$ where $p$ is the number of hyperparameters, and let the optimal hyperparameter combination $\alpha^*(T) \in A$, it holds that*

$$\mathbb{E}[R(T)] \leq \tilde{O}(T^{(p+2)/(p+3)}).$$

The detailed proof of Theorem 4.2 is presented in Appendix G. Note that this regret bound could be further improved to $\tilde{O}(T^{(p_0+2)/(p_0+3)})$ where $p_0$ is any constant that is no smaller than the zooming dimension of $(A, g_t), \forall t$. For example, from Figure 2 in Appendix A we can observe that in practice $g_t$ would be $C^2$-smooth and strongly concave, which implies that $\mathbb{E}[R(T)] \leq \tilde{O}(T^{(p+4)/(p+6)})$.

Note our work is the first one to consider model selection for bandits with a continuous candidate set, and the regret analysis for online model selection in the bandit setting Foster et al. (2019) is intrinsically difficult. For example, regret bound of the classic algorithm CORRAL Agarwal et al. (2017) is linearly dependent on the number of candidates and the regret of the worst model among them, which would be infinitely large in our case. And the non-stationarity under the *switching* environment would also deteriorate the optimal order of cumulative regret Cheung et al. (2019). Therefore, we believe our theoretical result is non-trivial and significant.

## 5 EXPERIMENTAL RESULTS

In this section, we show by experiments that our hyperparameter tuning framework outperforms the theoretical hyperparameter setting and other tuning methods with various (generalized) linear bandit algorithms. We utilize seven state-of-the-art bandit algorithms: two of them (LinUCB Li et al. (2010), LinTS Agrawal & Goyal (2013)) are linear bandits, and the other five algorithms (UCB-GLM Li et al. (2017), GLM-TSL Kveton et al. (2020), Laplace-TS Chapelle & Li (2011), GLOC Jun et al. (2017), SGD-TS Ding et al. (2021)) are GLBs. Note that all these bandit algorithms except Laplace-TS contain an exploration rate hyperparameter, while GLOC and SGD-TS further require an additional learning parameter. And Laplace-TS only depends on one stepsize hyperparameter for a gradient descent optimizer. We run the experiments on both simulations and the benchmark Movielens 100K dataset as well as the Yahoo News dataset. We compare our CDT framework with the theoretical setting, OP Bouneffouf & Claeys (2020) and TL Ding et al. (2022b) (one hyperparameter) and Syndicated Ding et al. (2022b) (multiple hyperparameters) algorithms. Due to space limit, the detailed descriptions of our experimental settings and the utilized algorithms, along with the results on the Yahoo News dataset, are deferred to Appendix A.4.1 and A.4.2. Since all the existing tuning algorithms require a user-defined candidate set, we design the tuning set for all potential hyperparameters as $\{0.1, 1, 2, 3, 4, 5\}$. And for our CDT framework, which is the first algorithm for tuning hyperparameters in an interval, we simply set the interval as $[0.1, 5]$ for all hyperparameters. Each experiment is repeated for 20 times, and the average regret curves with standard deviation are

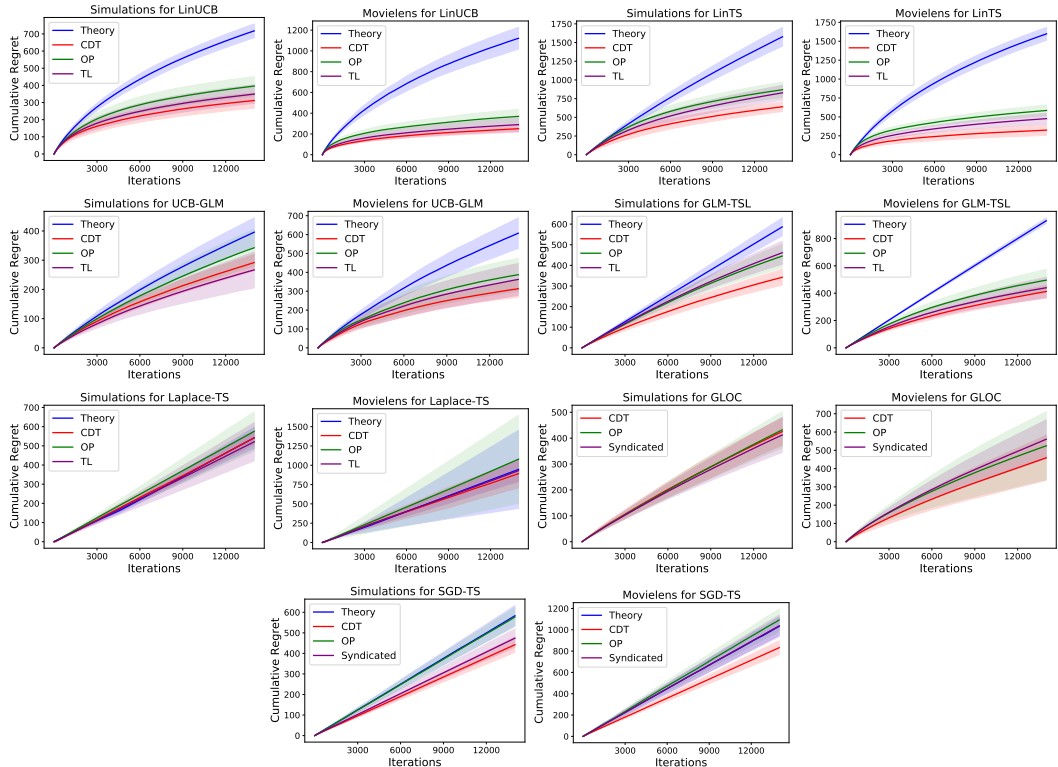

Figure 1: Cumulative regret curves of our CDT framework compared with existing hyperparameter selection methods under multiple (generalized) linear bandit algorithms.

displayed in Figure 1. We further explore the existing methods after enlarging the hyperparameter candidate set to validate the superiority of our proposed CDT in Appendix A.4.3.

We believe a large value of warm-up period $T_1$ may abandon some useful information in practice, and hence we use $T_1 = T^{2/(p+3)}$ according to Theorem 4.2 in experiments. And we would restart our hyperparameter tuning layer after every $T_2 = 3T^{(p+2)/(p+3)}$ rounds. An ablation study on the role of $T_1, T_2$ in our CDT framework is also conducted and deferred to Appendix A.4.4, where we demonstrate that the performance of CDT is pretty robust to the choice of $T_1, T_2$ in practice.

From Figure 1, we observe that our CDT framework outperforms all existing hyperparameter tuning methods for most contextual bandit algorithms. It is also clear that CDT performs stably and soundly with the smallest standard deviation across most datasets (e.g. experiments for LinTS, UCB-GLM), indicating that our method is highly flexible and robustly adaptive to different datasets. Moreover, when tuning multiple hyperparameters (GLOC, SGD-TS), we can see that the advantage of our CDT is also evident since our method is intrinsically designed for any hyperparameter space. It is also verified that the theoretical hyperparameter values are too conservative and would lead to terrible performance (e.g. LinUCB, LinTS). Note that all tuning methods exhibit similar results when applied to Laplace-TS. We believe it is because Laplace-TS only relies on an insensitive hyperparameter that controls the stepsize in gradient descent loops, which mostly affects the convergence speed.

## 6 CONCLUSION

In this paper, we propose the first online continuous hyperparameter optimization method for contextual bandit algorithms named CDT given the continuous hyperparameter search space. Our framework can attain sublinear regret bound in theory, and is general enough to handle the hyperparameter tuning task for most contextual bandit algorithms. Multiple synthetic and real experiments with multiple GLB algorithms validate the remarkable efficiency of our framework compared with existing methods in practice. In the meanwhile, we propose the Zooming TS algorithm with Restarts, which is the first work on Lipschitz bandits under the *switching* environment.

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
