## A   SUPPORTIVE EXPERIMENTAL DETAILS

Codes for our algorithms and experiments could be accessed using the following anonymous Github link: https://anonymous.4open.science/r/Continuous-Dynamic-Tuning/

### A.1   SIMULATIONS ON THE OPTIMAL HYPERPARAMETER VALUE IN GRID SEARCH

To further validate the necessity of dynamic hyperparameter tuning, we conduct a simulation for UCB algorithms LinUCB, UCB-GLM, GLOC and TS algorithms LinTS, GLM-TSL with a grid search of exploration parameter in $\{0.1, 0.5, 1, 1.5, 2, \ldots, 10\}$ and then report the best parameter value under different settings. Specifically, we set $d = 10, T = 8000, K = 60, 120$, and choose arm $x_{t,a}$ and $\theta^*$ randomly in $\{x : \|x\| \leq 1\}$. Rewards are simulated from $N(x_{t,a}^\top \theta^*, 0.5)$ for LinUCB, LinTS, and from Bernoulli$(1/(1 + \exp{(-x_{t,a}^\top \theta^*)}))$ for UCB-GLM, GLOC and GLM-TSL. The results are displayed in Table 1, where we can see that the optimal hyperparameter values are distinct and far from the theoretical ones under different algorithms or settings. Moreover, the theoretical optimal exploration rate should be identical under different values of $K$ for most algorithms shown here, but in practice the best hyperparameter to use depends on $K$, which also contradicts with the theoretical result.

| Bandit type | Linear bandit | | Generalized linear bandit | | |
|---|---|---|---|---|---|
| Algorithm | LinUCB | LinTS | UCB-GLM | GLOC | GLM-TSL |
| $K = 60$ | 2.5 | 1 | 1.5 | 4.5 | 1.5 |
| $K = 120$ | 3 | 1.5 | 2.5 | 5 | 2 |

Table 1: The optimal exploration parameter value in grid search for LinUCB, LinTS, UCB-GLM, GLOC and GLM-TSL based on average cumulative regret of 5 repeated simulations.

### A.2   SIMULATIONS TO VALIDATE THE LIPSCHITZNESS OF HYPERPARAMETER CONFIGURATION

We also conduct another simulation to show it is reasonable and fair to assume the expected reward is an almost-stationary Lipschitz function w.r.t. hyperparameter values. Specifically, we set $d = 6, T = 3000, K = 60$, and for each time we run LinUCB and LinTS by using our CDT framework, but also obtain the results by choosing the exploration hyperparameter in the set $\{0.3, 0.45, 0.6, \ldots, 8.85, 9\}$ respectively. For the first 200 rounds we use the random selection for sufficient exploration, and hence we omit the results for the first 200 rounds. After the warming-up period, we divide the rest of iterations into 140 groups uniformly, where each group contains 20 consecutive iterations. Then we calculate the mean of the obtained reward of each hyperparameter value in the adjacent 20 rounds, and centralize the mean reward across different hyperparameters in each group (we call it group mean reward). Afterward, we can calculate the mean and standard deviation of the group mean reward for different hyperparameter values across all groups. The results are shown in Figure 2, where we can see the group mean reward can be decently represented by a stationary Lipschitz continuous function w.r.t hyperparameter values. Conclusively, we could formulate the hyperparameter optimization problem as a stationary Lipschitz bandit after sufficient exploration in the long run. And in the very beginning we can safely believe there is also only finite number of change points. This fact firmly authenticates our problem setting and assumptions.

### A.3   SIMULATIONS FOR ALGORITHM 1

We also conduct empirical studies to evaluate our proposed Zooming TS algorithm with Restarts (Algorithm 1) in practice. Here we generate the dataset under the *switching* environment, and abruptly change the underlying mean function for several times within the time horizon $T$. The methods used for comparison as well as the simulation setting are elaborated as follows:

**Methods.** We compare our Algorithm 1 (we call it Zooming TS-R for abbreviation) with two contenders: (1) Zooming algorithm Kleinberg et al. (2019): this algorithm is designed for the static

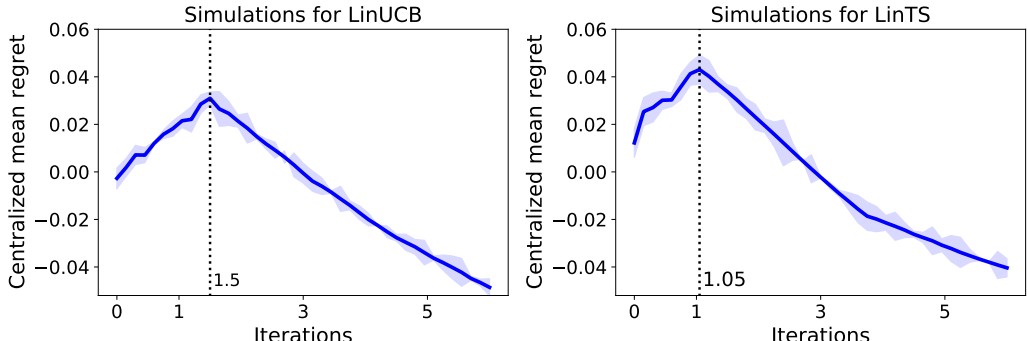

Figure 2: Average cumulative regret and its standard deviation of group mean reward for different hyperparameter values across all groups.

Lipschitz bandit, and would fail in theory under the *switching* environment; (2) Oracle: we assume this algorithm knows the exact time for all switching points, and would renew the Zooming algorithm when reaching a new stationary environment. Although this algorithm could naturally perform well, but it is infeasible in reality. Therefore, we would just use Oracle as a skyline here, and a direct comparison between Oracle and our Algorithm 1 is inappropriate.

**Settings.** Assume the set of arm is $[0, 1]$. The unknown mean function $f_t(x)$ is chosen from two classes of reward functions with different smoothness around their maximum: (1) $\{0.9 - 0.9|x - a|, x \in [0, 1] : a = 0.05, 0.25, 0.45, 0.70, 0.95\}$ (triangle function); (2) $\left\{\frac{2}{3\pi} \sin\left(\frac{3\pi}{2}(x - a + \frac{1}{3})\right), x \in [0, 1] : a = 0.05, 0.25, 0.45, 0.70, 0.95\right\}$. We set $T = 90,000$ and $c(T) = 3$, and choose the location of changing points at random in the very beginning. The random noise is generated according to $N(0, 0.1)$. The value of epoch size $H$ is set as suggested by our theory $H = 10\lceil(T/c(T))^{3/4}\rceil$. For each class of reward functions, we run the simulations for 20 times and report the average cumulative regret as well as the standard deviation for each contender in Figure 3. (The change points are fixed for each repetition to make the average value meaningful.)

Figure 3 shows the performance comparisons of three different methods under the *switching* environment measured by the average cumulative regret. We can see that Oracle is undoubtedly the best since it knows the exact times for all change points and hence restart our Zooming TS algorithm accordingly. The traditional Zooming algorithm ranks the last w.r.t both mean and standard deviation since it doesn't take the non-stationarity issue into account at all, and would definitely fail when the environment changes. This fact also coincides with our expectation precisely. Our proposed algorithm has an obvious advantage over the traditional Zooming algorithm when the change points exist, and we can see that our algorithm could adapt to the environment change quickly and smoothly.

## A.4 ADDITIONAL DETAILS AND RESULTS FOR SECTION 5

### A.4.1 EXPERIMENTAL SETTINGS FOR SECTION 5

Here we first summarize our proposed CDT method with other existing bandit hyperparameter tuning algorithms for comparison:

(1) **Theoretical setting**: We implement the theoretical exploration rate and stepsize for each algorithm. For the stepsize of gradient descent used in SGD-TS and Laplace-TS, we set it as 1 instead. (We observe the algorithmic performance is not sensitive to this stepsize.)

(2) **OP**: Bouneffouf & Claeys (2020) proposes OPLINUCB to tune the exploration rate of LinUCB. Here we modify it so that it could be used in other bandit algorithms. Note that OP is only applicable to algorithms with one hyperparameter, and hence we fix the learning parameter of GLOC and SGD-TS as their theoretical values instead, and only tune the exploration rates.

(3) **TL** Ding et al. (2022b) (one hyperparameter): For algorithms with only one hyperparameter, TL is used.

(4) **Syndicated** Ding et al. (2022b) (multiple hyperparameters): For GLOC and SGD-TS (two hyperparameters), the Syndicated framework is utilized for comparison.

Then we would present the details of our comparison settings regarding simulations, Movielens 100K dataset and YahooToday Module dataset in Section 5:

**Simulation**: In each repetition, we simulate all the feature vectors $\{x_{t,a}\}$ and the model parameter $\theta^*$ according to Uniform$(-1/\sqrt{r}, 1/\sqrt{r})$ elementwisely, and hence we have $\|x_{t,a}\| \leq 1$. We set $d = 25$, $K = 120$ and $T = 14,000$. For linear model, the expected reward of arm $a$ is formulated as $x_{t,a}^\top \theta^*$ and random noise is sampled from $N(0, 0.25)$; for Logistic model, the mean reward of arm $a$ is defined as $p = 1/(1 + \exp(-x_{t,a}^\top \theta^*))$, and the output is drawn from a Bernoulli distribution.

**Movielens 100K dataset**: This dataset contains 100K ratings from 943 users on 1,682 movies. For data pre-processing, we utilize LIBPMF Yu et al. (2014) to perform matrix factorization and obtain the feature matrices for both users and movies with $d = 20$, and then normalize all feature vectors into unit $r$-dimensional ball. In each repetition, the model parameter $\theta^*$ is defined as the average of 300 randomly chosen users' feature vectors. And for each time $t$, we randomly choose $K = 300$ movies from 1,682 available feature vectors as arms $\{x_{t,a}\}_{a=1}^{300}$. The time horizon $T$ is set to 14,000. For linear models, the expected reward of arm $a$ is formulated as $x_{t,a}^\top \theta^*$ and random noise is sampled from $N(0, 0.5)$; for Logistic model, the output of arm $a$ is drawn from the Bernoulli distribution with $p = 1/(1 + \exp(-x_{t,a}^\top \theta^*))$.

**Yahoo News dataset:** We downloaded the Yahoo Recommendation dataset R6A, which contains Yahoo data from May 1 to May 10, 2009 with $T = 2881$ timestamps. For each user's visit, the module will select one article from a pool of 20 articles for the user, and then the user will decide whether to click. We transform the contextual information into a 6-dimensional vector based on the processing in Chu et al. (2009). We build a Logistic bandit on this data, and the observed reward is simulated from a Bernoulli distribution with a probability of success equal to its click-through rate at each time.

### A.4.2 YAHOO NEWS RECOMMENDATION DATASET RESULTS

Since it is a logistic bandit, we only output the results of GLBs in the following Table 2:

| Method | UCB-GLM | GLM-TSL | Laplace-TS | GLOC | SGD-TS |
|---|---|---|---|---|---|
| Theory | 221.51 | 214.67 | 217.38 | | 206.73 |
| CDT | 221.69 | 218.27 | 217.05 | 217.95 | 218.35 |
| OP | 217.25 | 217.08 | 213.95 | 216.28 | 215.58 |
| TL/Syndicated | 218.95 | 219.36 | 214.42 | 218.19 | 215.02 |

Table 2: Comparisons of cumulative rewards from different algorithms on Yahoo dataset.

From the table above, we can see that our proposed CDT performs well on the Yahoo dataset. Specifically, it is only slightly worse than TL for GLM-TSL and GLOC, and yields the best results among all hyperparameter tuning frameworks for UCB-GLM, GLM-TSL, and SGD-TS. While the theoretical- hyperparameter setting slightly outperforms CDT in UCB-GLM and Laplace-TS, it is very unstable. And according to other experiments in Section 5, its cumulative regret will explode in large-scale experiments.

### A.4.3 BASELINES WITH A LARGE CANDIDATE SET

To further make a fair comparison and validate the high superiority of our proposed CDT framework over the existing OP, TL (or Syndicated) which relies on a user-defined hyperparameter candidate set, we explore whether CDT will consistently outperform if baselines are running with a large tuning set. Here we replace the original tuning set $C_1 = \{0.1, 1, 2, 3, 4, 5\}$ with a finer set $C_2 = \{0.1, 0.25, 0.5, 0.75, 1, 1.5, 2, 2.5, 3, 3.5, 4, 4.5, 5\}$. And the new results are shown in the following Table 3 (original results in Section 5 are in gray).

Therefore, we can observe that the performance overall becomes worse under $C_2$ compared with the original $C_1$. In other words, adding lots of elements to the tuning set will not help improve the performance of existing algorithms. We believe this is because the theoretical regret bound of TL (Syndicated) also depends on the number of candidates $k$ in terms of $\sqrt{k}$ Ding et al. (2022b). There

| Candidate Set | | C1 | | C2 | |
|---|---|---|---|---|---|
| Algorithm | Setting | TL/Syndicated | OP | TL/Syndicated | OP |
| LinUCB | Simulations | 343.14 | 383.62 | 356.23 | 389.91 |
| | Movielens | 346.16 | 390.10 | 359.10 | 408.67 |
| LinTS | Simulations | 828.41 | 869.30 | 874.34 | 925.29 |
| | Movielens | 519.09 | 666.35 | 516.62 | 667.77 |
| UCB-GLM | Simulations | 271.45 | 350.85 | 298.68 | 367.97 |
| | Movielens | 381.00 | 397.58 | 406.29 | 412.62 |
| GLM-TSL | Simulations | 433.27 | 445.43 | 448.21 | 458.71 |
| | Movielens | 446.74 | 678.91 | 458.23 | 718.46 |
| Laplace-TS | Simulations | 510.03 | 568.81 | 530.29 | 567.10 |
| | Movielens | 949.51 | 1063.92 | 958.10 | 1009.23 |
| GLOC | Simulations | 406.28 | 417.30 | 414.82 | 427.05 |
| | Movielens | 571.36 | 513.90 | 568.91 | 520.72 |
| SGD-TS | Simulations | 448.29 | 551.63 | 458.09 | 557.04 |
| | Movielens | 1016.72 | 1084.13 | 1038.94 | 1073.91 |

Table 3: Cumulative regrets of baselines under different hyperparameter tuning sets.

is no theoretical guarantee for OP. After introducing so many redundant values in the candidate set, the TL (Syndicated) and OP algorithms would get disturbed and waste lots of concentration on those unnecessary candidates.

In conclusion, we believe the existing algorithms relying on user-tuned candidate sets would perform well if the size of the candidate set is reasonable and the candidate set contains some value very close to the optimal hyperparameter value. However, in practice, finding the unknown optimal hyperparameter value is a black-box problem, and it's impossible to construct a candidate set satisfying the above requirements at the beginning. If we discretize the interval finely, then the large size of the candidate set would hurt the performance as well. On the other hand, our proposed CDT could adaptively "zoom in" on the regions containing this optimal hyperparameter value automatically, without the need of pre-specifying a "good" set of hyperparameters. And CDT could always yield robust results according to the extensive experiments we did in Section 5.

### A.4.4 ABLATION STUDY ON THE CHOICE OF $T_1$ AND $T_2$

For $T_1$, we set it to $T^{2/(p+3)}$ where $p$ stands for the number of hyperparameters according to Theorem 4.2. Specifically, for LinUCB, LinTS, UCB-GLM, GLM-TSL and Laplace-TS, we choose it to be 118. For GLOC and SGD-TS, we set it as 45. Here we also rerun our experiments in Section 5 with $T_1 = 0$ (no warm-up) since we believe a long warm-up period will abandon lots of useful information, and then we report the results after this change:

We can observe that the results are almost identical from Table 4. For $T_2$, Theorem 4.2 suggests that $T_2 = O\left(T^{(p+2)/(p+3)}\right)$. In our original experiments, we choose $T_2 = 3T^{(p+2)/(p+3)}$. To take an ablation study on $T_2$ we take $T_2 = kT^{(p+2)/(p+3)}$ for $k = 1, 2, 3$ in each experiment, and to see whether our CDT framework is robust to the choice of $k$.

According to Table 5, we can observe that overall $k = 2$ and $k = 3$ perform better than $k = 1$. We believe it is because, in the long run, the optimal hyperparameter would tend to be stable, and hence some restarts are unnecessary and inefficient. Note by choosing $k = 1$ our proposed CDT still outperforms the existing TL and OP tuning algorithms overall. For $k = 2$ and $k = 3$, we can observe that their performances are comparable, which implies that the choice of $k$ is quite robust in practice. We believe it is due to the fact that our proposed Zooming TS algorithm could always adaptively approximate the optimal point. Although it is unknown which one is better in practice under different cases, our comprehensive simulations show that choosing either one in practice will work well and outperform all the existing methods. In conclusion, these results suggest that we have a universal way to set the values of $T_1$ and $T_2$ according to the theoretical bounds, and we do not need to tune them for each particular dataset. In other words, the performance of our CDT tuning framework is robust to the choice of $T_1, T_2$ under different scenarios.

| Algorithm | Setting | $T_1 = 0$ | $T_1 = T^{2/(p+3)}$ |
|---|---|---|---|
| LinUCB | Simulation | 298.28 | 303.14 |
| | Movielens | 313.29 | 307.19 |
| LinTS | Simulation | 677.03 | 669.45 |
| | Movielens | 343.18 | 340.85 |
| UCB-GLM | Simulation | 299.74 | 300.54 |
| | Movielens | 314.41 | 311.72 |
| GLM-TSL | Simulation | 339.49 | 333.07 |
| | Movielens | 428.82 | 432.47 |
| Laplace-TS | Simulation | 520.29 | 520.35 |
| | Movielens | 903.16 | 900.10 |
| GLOC | Simulation | 414.70 | 418.05 |
| | Movielens | 455.39 | 461.78 |
| SGD-TS | Simulation | 430.05 | 425.98 |
| | Movielens | 843.91 | 838.06 |

Table 4: Ablation study on the role of $T_1$ in our CDT framework.

| Algorithm | Setting | $k = 1$ | $k = 2$ | $k = 3$ |
|---|---|---|---|---|
| LinUCB | Simulation | 328.28 | 300.62 | 298.28 |
| | Movielens | 310.06 | 303.10 | 313.29 |
| LinTS | Simulation | 717.77 | 670.90 | 677.03 |
| | Movielens | 360.12 | 352.19 | 343.18 |
| UCB-GLM | Simulation | 314.01 | 316.95 | 299.74 |
| | Movielens | 347.92 | 325.58 | 314.41 |
| GLM-TSL | Simulation | 320.21 | 331.43 | 339.49 |
| | Movielens | 439.98 | 428.91 | 428.82 |
| Laplace-TS | Simulation | 565.15 | 540.61 | 520.29 |
| | Movielens | 948.10 | 891.91 | 903.16 |
| GLOC | Simulation | 417.05 | 414.70 | 415.05 |
| | Movielens | 441.85 | 455.39 | 462.24 |
| SGD-TS | Simulation | 450.14 | 430.05 | 414.57 |
| | Movielens | 852.98 | 843.91 | 830.35 |

Table 5: Ablation study on the role of $T_2$ in our CDT framework.

### A.4.5 COMPUTATIONAL TIME

We also report the computational running time for the 10 cases corresponding to Figure 1 plots 1-10. Specifically, we display the average running time on each method in Table 6. We can observe that all existing methods and our CDT can run very fast in practice, and our CDT is only slightly more expensive in computation (CDT only takes at most two more seconds). In addition, we can conclude that the main computation time comes from the contextual bandit algorithm we tune on. We can see that GLM-TSL requires much more time than all other methods under different tuning methods. Therefore, we can conclude that our CDT significantly outperforms all existing baselines without increasing much computation.

## B  SUPPORTIVE REMARKS

*Remark* B.1.  (Can't use general bandit model selection methods) Although we acknowledge that our problem setting has some similarities with the model selection in bandits. Agarwal et al. (2017); Foster et al. (2019); Sharaf & Daumé III (2019), there are some significant discrepancies. To clarify this point, we take the state-of-the-art corralling idea Agarwal et al. (2017) as an example: in theory, it has regret bound or order $O(\sqrt{MT} + MR_{\max})$ where $M$ is the number of base models (number

| Algorithm | Setting | Theory | TL | OP | CDT |
|---|---|---|---|---|---|
| LinUCB | Simulation | 2.11 | 2.61 | 2.60 | 3.69 |
| | Movielens | 2.17 | 2.67 | 2.65 | 3.39 |
| LinTS | Simulation | 2.19 | 2.67 | 2.62 | 3.53 |
| | Movielens | 2.04 | 2.49 | 2.45 | 3.46 |
| UCB-GLM | Simulation | 8.34 | 8.84 | 8.71 | 9.57 |
| | Movielens | 7.89 | 8.44 | 8.35 | 9.47 |
| GLM-TSL | Simulation | 305.28 | 305.98s | 306.02 | 307.24 |
| | Movielens | 295.17 | 295.87 | 294.83 | 297.31 |
| Laplace-TS | Simulation | 525.62 | 526.31 | 526.24 | 527.75 |
| | Movielens | 502.89 | 503.45 | 503.51 | 504.45 |

Table 6: Running time for each method in seconds.

of hyperparameter combinations in our setting) and $R_{\max}$ is the regret of the **worst** candidate model in the tuning set. Therefore, on the one hand, $M$ is infinitely large in our problem setting with a continuous candidate set, which means the regret bound would also be infinitely large. On the other hand, in order to achieve sub-linear regret in hyperparameter tuning, the corralling idea requires that all hyperparameter candidates yield sub-linear regret in theory, which is a very unrealistic assumption. On the contrary, our work only assumes the existence of a hyperparameter candidate in the tuning set which yields good theoretical regret. In experiments, it is very expensive to use since it requires updating all base models at each round, and we have infinitely many base models under our setting. Moreover, Ding et al. (2022b) includes the corralling idea in their experiments, and we can observe that it performs terribly with almost linear regret in each setting. And this is because there is no sub-linear regret guarantee for it in the hyperparameter tuning problem. In conclusion, the only existing methods that really focus on hyperparameter tuning of bandits are OP and TL (Syndicated), and we use both of them in our paper as baselines.

*Remark* B.2. (Justifications on assumptions) We further explain the motivations of the Lipschitzness and piecewise stationarity assumptions of the expected reward function for hyperparameter tuning of bandit algorithms.

For Lipschitzness, we get the motivation of our formulation shown in Eqn. 3 and Eqn. 4 from the hyperparameter tuning work on the offline machine learning algorithms. Specifically, Bayesian optimization is widely considered as the state-of-the-art and most popular hyperparameter tuning method, which assumes that the underlying function is sampled from a Gaussian process in the given space. By selecting a value $x$ in the space and obtaining the corresponding reward, Bayesian optimization could update its estimation of the underlying function, especially in the neighbor of $x$ sequentially. And it also relies on a user-defined kernel function, whose selection is also purely empirical and lacks theoretical support. In our work, we use a similar idea as Bayesian optimization: close hyperparameters tend to yield similar values with other conditions fixed. And this natural extension motivates the Lipschitz assumption made in our paper. Therefore, it is fair to make a similar and analogous assumption (close hyperparameters yield similar results given other conditions fixed) for the hyperparameter tuning of bandit algorithms in our work. We validate this assumption using a suite of simulations in Appendix A.

For the piecewise stationarity, as we mention in Section 3, it is inappropriate to assume the strict stationarity of the bandit algorithm performance under the same hyperparameter value setting across time $T$. As an example, for most UCB and TS-based bandit algorithms (e.g. LinUCB, LinTS, UCB-GLM, GLM-UCB, GLM-TSL, etc.), the exploration degree of an arm is a multiplier of the exploration rate and the uncertainty of an arm. In the beginning, a moderate value of the exploration rate may lead to a large exploration degree for the arm since the uncertainty is large. On the contrary, in the long run, a moderate value of exploration rate will lead to a minor exploration degree for the arm since its value has been well estimated with small uncertainty. Therefore, a fixed hyperparameter setting may suggest different results across different stages of time, and hence it is unreasonable to expect the strong stationarity of the hyperparameter tuning for bandit algorithms at all time steps. On the other hand, it would be very inefficient to assume a completely non-stationary environment as in Ding et al. (2022b) which uses EXP3. In very close time steps, we could anticipate that the same hyperparameter setting would yield a very similar result in expectation since the uncertainty of any

arm would be close. And using a non-stationary environment will totally waste this information and hence is inefficient. Therefore, it is very well motivated to use a partial non-stationarity assumption that lies in the middle ground between the above two extremes. Note our proposed tuning method yields very promising results in extensive experiments under our formulations. And the stationary environment can be regarded as a special case of our switching environment setting where the functions in between all change points are the same.

Finally, we will explain why it is excessively difficult to present theoretical validation regarding these assumptions in our paper. As we mentioned, our formulation is motivated by Bayesian optimization, arguably the most popular method for hyperparameter tuning for offline machine learning algorithms. And we use a similar idea: similar hyperparameters tend to yield similar values while other conditions are fixed. However, people could hardly provide any theory backing for the analogous assumption of Bayesian optimization for any offline machine learning algorithms (e.g. regression, classification), and hyperparameter tuning is widely considered as a black-box problem for offline machine learning algorithms. Not to mention that the theoretical analysis of hyperparameter tuning for any bandit algorithm is much more challenging than that of offline machine learning algorithms since historical observations along with hyperparameter values will affect the online selection simultaneously for the bandit algorithms, and we can use different hyperparameters in different rounds for bandit algorithms. Conclusively, our formulation is natural and well-motivated.

## C    DETAILED PROOF ON THE ZOOMING DIMENSION

In the beginning, we would reload some notations for simplicity. Here we could omit the time subscript (or superscript) $t$ since the following result could be identically proved for each round $t$. Assume the Lipschitz function $f$ is defined on $\mathbb{R}^{p_c}$, and $v^* := \operatorname{argmax}_{v \in A} f(v)$ denotes the maximal point (w.l.o.g. assume it's unique), and $\Delta(v) = f(v^*) - f(v)$ is the "badness" of the arm $v$. We then naturally denote $A_r$ as the $r$-optimal region at the scale $r \in (0, 1]$, i.e. $A_r = \{v \in A : r/2 < \Delta(v) \leq r\}$. The $r$-zooming number could be denoted as $N_z(r)$. And the zooming dimension could be naturally denoted as $p_z$. Note that by the Assouad's embedding theorem, any compact doubling metric space $(A, \operatorname{Dist}(\cdot, \cdot))$ can be embedded into the Euclidean space with some type of metric. Therefore, for all compact doubling metric spaces with cover dimension $p_c$, it is sufficient to study on the metric space $([0, 1]^{p_c}, \|\cdot\|^l)$ for some $l \in (0, +\infty]$ instead.

We will rigorously prove the following two facts regarding the $r$-zooming number $N_z(r)$ of $(A, f)$ for arbitrary compact set $A \subseteq \mathbb{R}^{p_c}$ and Lipschitz function $f(\cdot)$ defined on $A$:

- $0 \leq p_z \leq p_c$.
- The zooming dimension could be much smaller than $p_c$ under some mild conditions. For example, if the payoff function $f$ is greater than $\|v^* - v\|^\beta$ in scale in a (non-trivial) neighborhood of $v^*$ for some $\beta \geq 1$, i.e. $f(v^*) - f(v) \geq C(\|v^* - v\|^\beta)$ as $\|v^* - v\| \leq r$ for some $C > 0$ and $r = \Theta(1)$, then it holds that $p_z \leq (1 - 1/\beta)p_c$. Note $\beta = 2$ when we have $f(\cdot)$ is $C^2$-smooth and strongly concave in a neighborhood of $v^*$, which subsequently implies that $p_z \leq p_c/2$.

*Proof.* Due to the compactness of $A$, it suffices to prove the results when $A = [0, 1]^{p_c}$. By the definition of the zooming dimension $p_z$, it naturally holds that $p_z \geq 0$. On the other side, since the space $A$ is a closed and bounded set in $\mathbb{R}^{p_c}$, we assume the radius of $A$ is no more than $S$, which consequently implies that the $r/16$-covering number of $A$ is at most the order of

$$\left(\frac{S}{\frac{r}{16}}\right)^{p_c} = (16S)^{p_c} \cdot r^{-p_c}.$$

Since we know $A_r \subseteq A$, it holds that $p_z \leq p$. Secondly, if the payoff function $f$ is locally greater than $\|v^* - v\|^\beta$ in scale for some $\beta \geq 1$, i.e. $f(v^*) - f(v) \geq C(\|v^* - v\|^\beta)$, then there exists $C \in \mathbb{R}$ and $\delta > 0$ such that as long as $C \|v - v^*\|^\beta \leq \delta$ we have $f(v^*) - f(v) \geq C \|v - v^*\|^\beta$. Therefore, for $0 < r < \delta$, it holds that,

$$\{v : r \geq f(v^*) - f(v) > r/2\} \subseteq \{v : C \|v - v^*\|^\beta \leq r\} = \left\{v : \|v - v^*\| \leq \left(\frac{r}{C}\right)^{\frac{1}{\beta}}\right\}$$

Figure 3: Cumulative regret plots of Zooming TS-R, Zooming and Oracle algorithms under the *switching* environment.

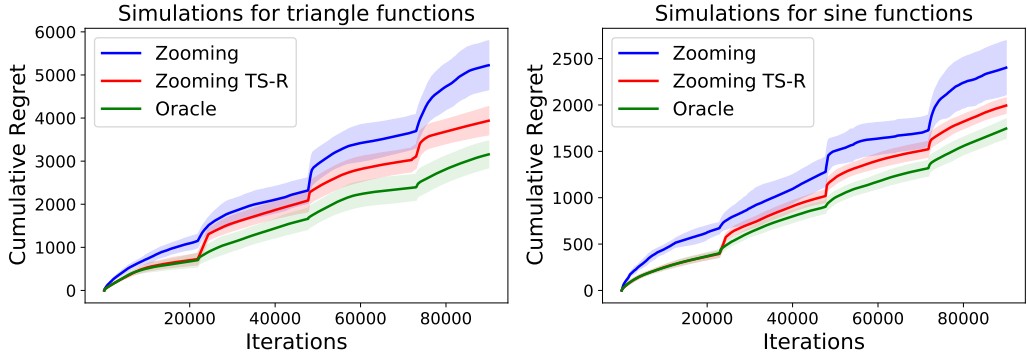

It holds that the $r$-covering number of the Euclidean ball with center $v^*$ and radius $(r/c)^{(1/\beta)}$ is of the order of

$$\left( \frac{\left( \frac{r}{C} \right)^{\frac{1}{\beta}}}{\frac{r}{16}} \right)^{p_c} = \left( \frac{16}{C^{\frac{1}{\beta}}} \right)^{p_c} \cdot r^{-(1 - \frac{1}{\beta})p_c}$$

which explicitly implies that $p_z \leq (1 - 1/\beta)p_c$. $\qquad\square$

## D   INTUITION OF OUR THOMPSON SAMPLING UPDATE

Intuitively, we consider a Gaussian likelihood function and Gaussian conjugate prior to design our Thompson Sampling version of zooming algorithm, and here we would ignore the clipping step for explanation. Suppose the likelihood of reward $\tilde{y}_t$ at time $t$, given the mean of reward $I(v_t)$ for our pulled arm $v_t$, follows a Gaussian distribution $N(I(v_t), s_0^2)$. Then, if the prior of $I(v_t)$ at time $t$ is given by $N(\hat{f}_t(v_t), s_0^2/n_t(v_t))$, we could easily compute the posterior distribution at time $t+1$,

$$\Pr(I(v_t)|\tilde{y}_t) \propto \Pr(\tilde{y}_t|I(v_t))\Pr(I(v_t)),$$

as $N(\hat{f}_{t+1}(v_t), s_0^2/n_{t+1}(v_t))$. We can see this result coincides with our design in Algorithm 1 and its proof is as follows:

*Proof.*

$$\Pr(I(v_t)|\tilde{y}_t) \propto \Pr(\tilde{y}_t|I(v_t))\Pr(I(v_t))$$

$$\propto \exp\left\{ -\frac{1}{2s_0^2}[(I(v_t) - \tilde{y}_t)^2 + n_t(v_t)(I(v_t) - f_t(v_t))^2] \right\}$$

$$\propto \exp\left\{ -\frac{1}{2s_0^2}[(n_t(v_t) + 1)I(v_t)^2 - 2(\tilde{y}_t + n_t(v_t)f_t(v_t))I(v_t)] \right\}$$

$$\propto \exp\left\{ -\frac{n_{t+1}(v_t)}{2s_0^2}\left[ I(v_t)^2 - 2\frac{(\tilde{y}_t + n_t(v_t)f_t(v_t))}{n_{t+1}(v_t)}I(v_t) \right] \right\}$$

$$\propto \exp\left\{ -\frac{n_{t+1}(v_t)}{2s_0^2}(I(v_t) - f_{t+1}(v_t))^2 \right\}$$

Therefore, the posterior distribution of $I(v_t)$ at time $t+1$ is $N(f_{t+1}(v_t), s_0^2 \frac{1}{n_{t+1}(v_t)})$. $\qquad\square$

This gives us an intuitive explanation why our Zooming TS algorithm works well when we ignore the clipped distribution step. And we have stated that this clipping step is inevitable in Lipschitz bandit setting in our main paper since (1) we'd like to avoid underestimation of good active arms, i.e. avoid the case when their posterior samples are too small. (2) We could at most adaptively zoom in the regions which contains $v^*$ instead of exactly detecting $v^*$, and this inevitable loss could be mitigated by setting a lower bound for TS posterior samples. Note that although the intuition of our Zooming

TS algorithm comes from the case where contextual bandit rewards follow a Gaussian distribution, we also prove that our algorithm can achieve a decent regret bound under the *switching* environment and the optimal instance-dependent regret bound under the stationary Lipschitz bandit setting.

# E  PROOF OF THEOREM 4.1

## E.1  STATIONARY ENVIRONMENT CASE

To prove Theorem 4.1, we will first focus on the stationary case, where $f_t := f, \ \forall t \in [T]$. When the environment is stationary, we could omit the subscript (or superscript) $t$ in some notations as in Section C for simplicity: Assume the Lipschitz function is $f$, and $v^* := \text{argmax}_{v \in A} f(v)$ denotes the maximal point (w.l.o.g. assume it's unique), and $\Delta(v) = f(v^*) - f(v)$ is the "badness" of the arm $v$. We then naturally denote $A_r$ as the $r$-optimal region at the scale $r \in (0, 1]$, i.e. $A_r = \{v \in A : r/2 < \Delta(v) \leq r\}$. The $r$-zooming number could be denoted as $N_z(r)$. And the zooming dimension could be naturally denoted as $p_z$. Note we could omit the subscript (or superscript) $t$ for the notations just mentioned above since all these values would be fixed through all rounds under the stationary environment.

### E.1.1  USEFUL LEMMAS AND COROLLARIES

Recall that $\hat{f}_t(v)$ is the average observed reward for arm $v \in A$ by time $t$. And we call all the observations (pulled arms and observed rewards) over $T$ total rounds as a process.

**Definition E.1.** We call it a clean process, if for each time $t \in [T]$ and each strategy $v \in A$ that has been played at least once at any time $t$, we have $|\hat{f}_t(v) - f(v)| \leq r_t(v)$.

**Lemma E.2.** *The probability that, a process is clean, is at least $1 - 1/T$.*

*Proof.* Fix some arm $v$. Recall that each time an algorithm plays arm $v$, the reward is sampled IID from some distribution $\mathbb{P}_v$. Define random variables $U_{v,s}$ for $1 \leq s \leq T$ as follows: for $s \leq n_T(v)$, $U_{v,s}$ is the reward from the $s$-th time arm $v$ is played, and for $s > n_T(v)$ it is an independent sample from $\mathbb{P}_v$. For each $k \leq T$ we can apply Chernoff bounds to $\{U_{v,s} : 1 \leq s \leq k\}$ and obtain that:

$$\Pr\left(\left|\frac{1}{k}\sum_{s=1}^{k} U_{v,s} - f(v)\right| \geq \sqrt{\frac{13\tau_0^2 \ln T}{2k}}\right) \leq 2 \cdot \exp\left(-\frac{k}{2\tau_0^2}\frac{13\tau_0^2 \ln T}{2k}\right)$$

$$= 2\exp\left(\frac{13}{4}\ln T\right) = 2T^{-3.25} \leq T^{-3}, \tag{7}$$

since we can trivially assume that $T \geq 16$. Let $N$ be the number of arms activated all over rounds $T$; note that $N \leq T$. Define $X$-valued random variables $\{x_i\}_{i=1}^T$ as follows: $x_j$ is the $\min(j, N)$-th arm activated by time $T$. For any $x \in A$ and $j \leq T$, the event $\{x = x_j\}$ is independent of the random variables $\{U_{x,s}\}$: the former event depends only on payoffs observed before $x$ is activated, while the latter set of random variables has no dependence on payoffs of arms other than $x$. Therefore, Eqn. (7) is still valid if we replace the probability on the left side with conditional probability, conditioned on the event $\{x = x_j\}$. Taking the union bound over all $k \leq T$, it follows that:

$$\Pr(\forall t \leq T, |f(v) - \hat{f}_t(v)| \leq r_t(v) \,|\, x_j = v) \geq 1 - T^{-2}, \quad \forall v \in A, j \in [T],$$

Integrating over all arms $v$ we get

$$\Pr(\forall t \leq T, |f(x_j) - \hat{f}_t(x_j)| \leq r_t(x_j)) \geq 1 - T^{-2}, \quad \forall j \in [T].$$

Finally, we take the union bound over all $j \leq T$, and it holds that,

$$\Pr(\forall t \leq T, j \leq T, |f(x_j) - \hat{f}_t(x_j)| \leq r_t(x_j)) \geq 1 - T^{-1},$$

and this obviously implies the result. $\qquad\square$

**Lemma E.3.** *If it is a clean process, then $B(v, r_t(v))$ could never be eliminated from Algorithm 1 for any $t \in [T]$ and arm $v$ that is active at round $t$, given that $v^* \in B(v, r_t(v))$.*

*Proof.* Recall that from Algorithm 1, at round $t$ the ball $B(u, r_t(u))$ would be permanently removed if we have for some active arm $v$ s.t.

$$\hat{f}_t(v) - r_t(v) > \hat{f}_t(u) + 2r_t(u).$$

If we have that $v^* = \text{argmax}_{x \in A} f(x) \in B(u, r_t(u))$, then it holds that

$$\hat{f}_t(u) + 2r_t(u) \geq f(u) + r_t(u) \geq f(u) + \text{Dist}(u, v^*) \geq f(v^*),$$

where the first inequality is due to the clean process and the last one comes from the fact that $f$ is a Lipschitz function. On the other hand, we have that for any active arm $v$,

$$f(v) \geq \hat{f}_t(v) - r_t(v), \quad f(v^*) \geq f(v).$$

Therefore, it holds that

$$\hat{f}_t(v) - r_t(v) \leq \hat{f}_t(u) + 2r_t(u).$$

And this inequality concludes our proof. $\qquad\square$

**Lemma E.4.** *If it is a clean process, then for any time $t$ and any active strategy $v$ that has been played at least once before time $t$ we have $\Delta(v) \leq 5\mathbb{E}[r_t(v)]$. Furthermore, it holds that $\mathbb{E}(n_t(v)) \leq O(\ln(T)/\Delta(v)^2)$.*

*Proof.* Let $S_t$ be the set of all arms that are active at time $t$. Suppose an arm $v_t$ is played at time $t$ and was previously played at least twice before time $t$. Firstly, We would claim that

$$f(v^*) \leq I_t(v_t) \leq f(v_t) + 3r_t(v_t)$$

holds uniformly for all $t$ with probability at least $1 - \delta$, which directly implies that $\Delta(v_t) \leq 3r_t(v_t)$ with high probability uniformly. First we show that $I_t(v_t) \geq f(v^*)$. Indeed, recall that all arms are covered at time $t$, so there exists an active arm $v_t^*$ that covers $v^*$, meaning that $v^*$ is contained in the confidence ball of $v_t^*$. And based on Lemma E.3 the confidence ball containing $v^*$ could never be eliminated at round $t$ when it's a clean process. Recall $Z_{t,v}$ is the i.i.d. standard normal random variable used for any arm $v$ in round $t$ (Eqn. (6)). Since arm $v_t$ was chosen over $v_t^*$, we have $I_t(v_t) \geq I_t(v_t^*)$. Since this is a clean process, it follows that

$$I_t(v_t^*) = \hat{f}_t(v_t^*) + s_0\sqrt{\frac{1}{n_t(v_t^*)}}Z_{t,v_t^*} \geq f(v_t^*) + s_0\sqrt{\frac{1}{n_t(v_t^*)}}Z_{t,v_t^*} - r_t(v_t^*) \tag{8}$$

Furthermore, according to the Lipschitz property we have

$$f(v_t^*) \geq f(v^*) - \text{Dist}(v_t^*, v^*) \geq f(v^*) - r_t(v_t^*). \tag{9}$$

Combine Eqn. (8) and (9), we have

$$I_t(v_t) \geq I_t(v_t^*) \geq f(v^*) + s_0\sqrt{\frac{1}{n_t(v_t^*)}}Z_{t,v_t^*} - 2r_t(v_t^*)$$

$$= f(v^*) + \sqrt{\frac{52\pi\tau_0^2\ln(T)}{n_t(v_t^*)}}\left(Z_{t,v_t^*} - \frac{1}{\sqrt{2\pi}}\right) \geq f(v^*), \tag{10}$$

where we get the last inequality since we truncate the random variable $Z_{t,v_t^*}$ by the lower bound $1/\sqrt{2\pi}$ according to the definition. On the other hand, we have

$$I_t(v_t) \leq f(v_t) + r_t(v_t) + s_0\sqrt{\frac{1}{n_t(v_t)}}Z_{t,v_t} = f(v_t) + \left(1 + 2\sqrt{2\pi}Z_{t,v_t}\right)r_t(v_t) \tag{11}$$

Therefore, by combing Eqn. (10) and (11) we have that

$$\Delta(v_t) \leq \left(1 + 2\sqrt{2\pi}Z_{t,v_t}\right)r_t(v_t). \tag{12}$$

And we know that $Z_{t,:}$ is defined as $Z_{t,:} = \max\{1/\sqrt{2\pi}, \tilde{Z}_{t,:}\}$ where $\tilde{Z}_{t,:}$ is iid drawn from standard normal distribution. In other words, $Z_{t,v_t}$ follows a clipped normal distribution with the following PDF:

$$f(x) = \begin{cases} \phi(x) + (1 - \Phi(x))\delta\left(x - \frac{1}{\sqrt{2\pi}}\right), & x \geq \frac{1}{\sqrt{2\pi}}; \\ 0, & x < \frac{1}{\sqrt{2\pi}}; \end{cases}$$

Here $\phi(\cdot)$ and $\Phi(\cdot)$ denote the PDF and CDF of standard normal distribution. And we have

$$\mathbb{E}(Z_{t,v_t}) \leq \frac{1}{\sqrt{2\pi}} + \int_{\frac{1}{\sqrt{2\pi}}}^{+\infty} x\phi(x)dx \leq \frac{1}{\sqrt{2\pi}} + \frac{1}{\sqrt{2\pi}}e^{-\frac{1}{4\pi}} \leq \sqrt{\frac{2}{\pi}}$$

By taking expectation on Eqn. (12), we have $\Delta(v_t) \leq 5\mathbb{E}(r_t(v_t))$. Next, we would show that $\mathbb{E}(n_t(v_t)) \leq O(\ln(T))/\Delta(v_t)^2$. Based on Eqn. (11) and the definition of $r_t(\cdot)$, we could deduce that

$$\sqrt{n_t(v_t)} \leq \sqrt{\frac{13}{2}\tau_0^2 \ln(T)}(1 + 2\sqrt{2\pi}Z_{t,v_t})\frac{1}{\Delta(v_t)},$$

which thus implies that

$$n_t(v_t) \leq \frac{13}{2}\tau_0^2 \ln(T)(1 + 2\sqrt{2\pi}Z_{t,v_t})^2 \frac{1}{\Delta(v_t)^2} = O(\ln(T))(1 + 2\sqrt{2\pi}Z_{t,v_t})^2 \frac{1}{\Delta(v_t)^2}. \quad (13)$$

By simple calculation, we could show that

$$\mathbb{E}(Z_{t,v_t}^2) \leq \frac{1}{2\pi} + \int_{\frac{1}{\sqrt{2\pi}}}^{+\infty} x^2\phi(x)dx \leq \frac{1}{\pi} + \frac{1}{2} \leq 1$$

$$\Rightarrow \quad \mathbb{E}\left[(1 + 2\sqrt{2\pi}Z_{t,v_t})^2\right] \leq 1 + 4\sqrt{2\pi}\sqrt{\frac{2}{\pi}} + 8\pi < +\infty.$$

After revisiting Eqn. (13), we can show that $\mathbb{E}(n_t(v_t)) \leq O(\ln(T))/\Delta(v_t)^2$. Now suppose arm $v$ is only played once at time $t$, then $r_t(v) > 1$ and thus the lemma naturally holds. Otherwise, let $s$ be the last time arm $v$ has been played according to the selection rule, where we have $r_t(v) = r_s(v)$, and then based on Eqn. (11) it holds that

$$I_t(v) \leq f(v) + \left(1 + 2\sqrt{2\pi}Z_{s,v}\right)r_t(v).$$

And then we could show that $\Delta(v) \leq 5\mathbb{E}(r_t(v))$. By using an identical argument as before, we could show that $\mathbb{E}(n_t(v)) \leq O(\ln(T))/\Delta(v)^2$. □

**Lemma E.5.** *Let $X_1, \ldots, X_n$ be independent $\sigma^2$-sub-Gaussian random variables. Then for every $t > 0$,*

$$P\left(\max_{1,\leq,n} X_i \geq \sqrt{2\sigma^2(\ln(T) + t)}\right) \leq e^{-t}.$$

*Proof.* Let $u = \sqrt{2\sigma^2(\ln(n) + t)}$, we have

$$P\left(\max_{1,\leq,n} X_i \geq u\right) = P(\exists i, X_i \geq u) \leq \sum_{i=1}^{n} P(X_i \geq u) \leq ne^{-\frac{u^2}{2\sigma^2}} = e^{-t}.$$

□

### E.1.2 PROOF OF THEOREM 4.1 UNDER STATIONARY ENVIRONMENT

*Proof.* By Lemma E.2 we know that it is a clean process with probability at least $1 - \frac{1}{T}$. In other words, denote the event $\Omega := \{\text{clean process}\}$, and then we have that $P(\Omega) \geq 1 - \frac{1}{T}$. And according to Lemma E.3 we're aware that the active confidence balls containing the best arm can't be removed in a clean process. Remember that we use $S_T$ as the set of all arms that are active in the end, and denote

$$B_{i,T} = \left\{v \in S_T : 2^i \leq \frac{1}{\Delta(v)} < 2^{i+1}\right\}, \quad \text{where } S_T = \bigcup_{i=0}^{+\infty} B_{i,T},$$

where $i \geq 0$. Then, under the event $\Omega$, by using Corollary E.4 we have $\mathbb{E}(n_T(v)|\Omega) \leq O(\ln T)/\Delta(v)^2$, and hence it holds that

$$\sum_{v \in B_{i,T}} \Delta(v)\mathbb{E}(n_T(v)|\Omega) \leq O(\ln T) \sum_{v \in B_{i,t}} \frac{1}{\Delta(v)} \leq O(\ln T) \cdot 2^i |B_{i,t}|$$

Denote $r_i = 2^{-i}$, we have

$$\sum_{v \in B_{i,T}} \Delta(v)\mathbb{E}(n_T(v)|\Omega) \leq O(\ln T) \cdot \frac{1}{r_i}|B_{i,t}|$$

Next, we would show that for any active arms $u, v$ we have

$$\text{Dist}(u,v) > \frac{1}{4\sqrt{2\pi \ln(T)}} \min\{\Delta(u), \Delta(v)\} \tag{14}$$

with probability at least $1 - \frac{1}{T}$. W.l.o.g assume $u$ has been activated before $v$. Let $s$ be the time when $v$ has been activated. Then by the philosophy of our algorithm we have that $\text{Dist}(u,v) > r_s(v)$. Then according to Eqn. (12) in the proof Lemma E.4, it holds that $r_s(v) \geq \frac{1}{2\sqrt{2\pi}Z}\Delta(v)$ for some random variable $Z$ following the clipped standard normal distribution. Define the event $\Upsilon = \{Z_{t,v_t} < 2\sqrt{\ln(T)}$ for all $t \in [T]\}$, then based on Lemma E.5 we have $P(\Upsilon) \geq 1 - \frac{1}{T}$. Then under the event $\Upsilon$, we have $r_s(v) \geq \frac{1}{4\sqrt{2\pi \ln(T)}}\Delta(v)$, which then implies that Eqn. (14) holds under $\Upsilon$. Since for arbitrary $x, y \in B_{i,T}$ we have

$$\frac{r_i}{2} < \Delta(x) \leq r_i, \quad \frac{r_i}{2} < \Delta(y) \leq r_i,$$

which implies that under the event $\Upsilon$

$$\text{Dist}(x,y) > \frac{1}{4\sqrt{2\pi \ln(T)}} \min\{\Delta(x), \Delta(y)\} > \frac{r_i}{8\sqrt{2\pi \ln(T)}}.$$

Therefore, $x$ and $y$ should belong to different sets of $(r_i/8\sqrt{2\pi \ln(T)})$-diameter-covering. It follows that $|B_{i,T}| \leq N_z(r_i/8\sqrt{2\pi \ln(T)}) \leq O(\ln(T)^p)cr_i^{p_z} \leq \tilde{O}(cr_i^{p_z})$. Recall $N_z(r)$ is defined as the minimal number of balls of radius no more than $r$ required to cover $A_r$. As a result, under the events $\Omega$ and $\Upsilon$, it holds that

$$\sum_{v \in B_{i,T}} \Delta(v)\mathbb{E}(n_T(v) \,|\, \Omega \cap \Upsilon) \leq O(\ln T) \cdot \frac{1}{r_i}N_z(r_i) \tag{15}$$

Therefore, based on Eqn. (15), we have

$$R_L(T) = \sum_{v \in S_T} \Delta(v)\mathbb{E}(n_T(v))$$

$$= P(\Omega \cap \Upsilon) \sum_{v \in S_T} \Delta(v)\mathbb{E}(n_T(v) \,|\, \Omega \cap \Upsilon) + P(\Omega^c \cup \Upsilon^c) \sum_{v \in S_T} \Delta(v)\mathbb{E}(n_T(v) \,|\, \Omega^c \cup \Upsilon^c)$$

$$\leq \sum_{v \in S_T : \Delta(v) \leq \rho} \Delta(v)\mathbb{E}(n_T(v) \,|\, \Omega \cap \Upsilon) + \sum_{v \in S_T : \Delta(v) > \rho} \Delta(v)\mathbb{E}(n_T(v) \,|\, \Omega \cap \Upsilon) + \frac{2}{T} \cdot T$$

$$\leq \rho T + \sum_{i < \log_2(\frac{1}{\rho})} \frac{1}{r_i}\tilde{O}(cr_i^{-p_z}) + 2$$

$$\leq \rho T + \tilde{O}(1) \sum_{i < \log_2(\frac{1}{\rho})} \frac{1}{r_i}cr_i^{-p_z} + 2$$

$$\leq \rho T + \tilde{O}(1) \sum_{k=0}^{\lfloor \log_{1/2} 2\rho \rfloor} c2^{k(p_z+1)} + 2$$

$$\leq \rho T + \tilde{O}(1) \cdot 2 \cdot 2^{\lfloor \log_{1/2} 2\rho \rfloor(p_z+1)} + 2$$

$$\leq \rho T + \tilde{O}(1)\left(\frac{1}{2\rho}\right)^{p_z+1} + 2$$

By choosing $\rho$ in the scale of

$$\rho = \tilde{O}\left(\frac{1}{T}\right)^{\frac{1}{p_z+2}},$$

it holds that

$$R_L(T) = \tilde{O}\left(T^{\frac{p_z+1}{p_z+2}}\right).$$

### E.2 *Switching* (Non-stationary) Environment Case

Since there are $c(T)$ change points for the environment Lipschitz functions $f_t(\cdot)$, i.e.

$$\sum_{t=1}^{T-1} \mathbf{1}[\exists m \in A \,:\, f_t(m) \neq f_{t+1}(m)] = c(T).$$

Given the length of epochs as $H$, we would have $\lceil T/H \rceil$ epochs overall. And we know that among these $\lceil T/H \rceil$ different epochs, at most $c(T)$ of them contain the change points. For the rest of epochs that are free of change points, the cumulative regret could be bounded by the result we just deduced for the stationary case above. And the cumulative regret in any epoch with stationary environment could be bounded as $H^{(p_{z,*}+1)/(p_{z,*}+2)}$. Specifically, we could partition the $T$ rounds into $m = \lceil T/H \rceil$ epochs:

$$[T_1 + 1, T] = [\omega_0 = T_1 + 1, \omega_1) \cup [\omega_1, \omega_2) \cup \cdots \cup [\omega_{m-1}, \omega_m = T + 1),$$

where $\omega_{i+1} = \omega_i + H$ for $i = 0, \ldots, m-2$. Denote all the change points as $T_1 \leq \rho_1 < \cdots < \rho_{c(T)} \leq T$, and then define

$$\Omega = \{\cup[\omega_i, \omega_{i+1}) \,:\, \rho_j \in [\omega_i, \omega_{i+1}), \exists j = 1, \ldots c; \, i = 0, \ldots, m-1\}.$$

Then it holds that $|\Omega| \leq Hc(T)$. Therefore, it holds that

$$R_L(T) \leq \tilde{O}\left(Hc(T) + \left(\frac{T}{H} + 1\right) H^{\frac{p_{z,*}+1}{p_{z,*}+2}}\right) \leq \tilde{O}\left(Hc(T) + \frac{T}{H} \cdot H^{\frac{p_{z,*}+1}{p_{z,*}+2}}\right),$$

where the first part bound the regret of non-stationary epochs and the second part bound that of stationary ones. By taking $H = (T/c(T))^{(p_{z,*}+2)/(p_{z,*}+3)}$, it holds that

$$R_L(T) \leq \tilde{O}\left(T^{\frac{p_{z,*}+2}{p_{z,*}+3}} c(T)^{\frac{1}{p_{z,*}+3}}\right).$$

And this concludes our proof for Theorem 4.1. $\qquad\square$

## F Algorithm 1 with unknown $c(T)$ and $p_{z,*}$

### F.1 Introduction of Algorithm 3

When both the number of change points $c(T)$ over the total time horizon $T$ and the zooming dimension $p_{z,*}$ are unknown, we could adapt the BOB idea used in Cheung et al. (2019); Zhao et al. (2020) to choose the optimal epoch size $H$ based on the EXP3 meta algorithm. In the following, we first describe how to use the EXP3 algorithm to choose the epoch size dynamically even if $c(T)$ and $p_{z,*}$ are unknown. Then we present the regret analysis in Theorem F.1 and its proof.

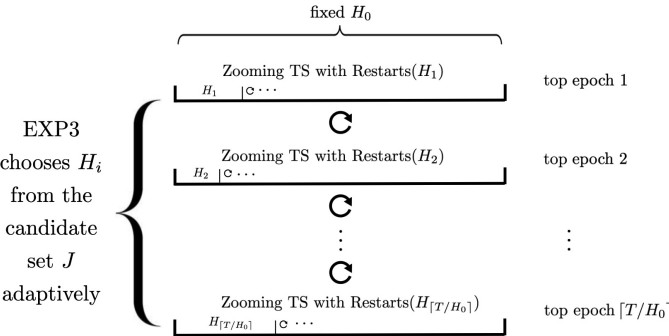

Figure 4: An illustration of Zooming TS algorithm with double restarts when $c(T)$ is agnostic.

Although the zooming dimension $p_{z,*}$ is unknown, it holds that $p_{z,*} \leq p_c$, and hence we could simply use the upper bound of $p_{z,*}$ (denoted as $p_u$) as $p_c$ instead (recall $p_c$ is the covering dimension).

Note that the upper bound $p_{z,*}$ could be more specific when we have some prior knowledge of the reward Lipschitz function $f(\cdot)$: for example, as we mentioned in Appendix C, if the function $f(\cdot)$ is known to be $C^2-$smooth and strongly concave in a neighborhood of its maximum defined in $\mathbb{R}^{p_c}$, it holds that $p_{z,*} \leq p_c/2$ and then we could use $p_u = p_c/2$ as the upper bound. Note that we also use the BOB mechanism in the CDT framework for hyperparameter tuning in Algorithm 2, where we treat the zooming TS algorithm with Restarts as the meta algorithm to select the hyperparameter setting in the upper layer, and then use the selected configuration for the bandit algorithm in the lower layer. However, here we would use BOB mechanism differently: we firstly divide the total horizon $T$ into several epochs of the same length $H_0$ (named top epoch), where in each top epoch we would restart the Algorithm 1. And in the $i-$th top epoch the restarting length $H_i$ (named bottom epoch) of Algorithm 1 could be chosen from the set $J = \{J_i := \lceil k \rceil : k \geq 1, k = H_0/2^{i-1}, i = 1, 2, \dots \}$, where the chosen bottom epoch size could be adaptively tuned by using EXP3 as the meta algorithm. Here we restart the zooming TS algorithm from two perspectives, where we first restart the zooming TS algorithm with Restarts (Algorithm 1) in each top epoch of some fixed length $H_0$, and then for each top epoch the restarting length $H_i$ for Algorithm 1 would be tuned on the fly based on the previous observations Cheung et al. (2019). Therefore, we would name this method Zooming TS algorithm with Double Restarts.

As for how to choose the bottom epoch size $H_i$ in each top epoch of length $H_0$, we implement a two-layer framework: In the upper layer, we use the adversarial MAB algorithm EXP3 to pull the candidate from $J = \{J_i\}$. And then in the lower layer we use it as the bottom epoch size for Algorithm 1. When a top epoch ends, we would update the components in EXP3 based on the rewards witnessed in this top epoch. The illustration of this double restarted strategy is depicted in Figure 4. And the detailed procedure is shown in Algorithm 3.

**Theorem F.1.** *By using the (top) epoch size as $H_0 = \lceil T^{(p_u+2)/(p_u+4)} \rceil$, the expected total regret of our Zooming TS algorithm with Double Restarts (Algorithm 3) under the switching environment over time $T$ could be bounded as*

$$R_L(T) \leq \tilde{O}\left( T^{\frac{p_u+2}{p_u+3}} \cdot \max\left\{ c(T)^{\frac{1}{p_u+3}}, T^{\frac{1}{(p_u+3)(p_u+4)}} \right\} \right).$$

*Specifically, it holds that*

$$R_L(T) \leq \begin{cases} T^{\frac{p_u+2}{p_u+3}} c(T)^{\frac{1}{p_u+3}}, & c(T) \geq T^{\frac{1}{p_u+4}}, \\ T^{\frac{p_u+3}{p_u+4}}, & c(T) < T^{\frac{1}{p_u+4}}, \end{cases}$$

*where $p_u \leq p_c$ is the upper bound of $p_{z,*}$.*

Therefore, we observe that if $c(T)$ is large enough, we could obtain the same regret bound as in Theorem 4.1 given $p_{z,*}$.

## F.2 PROOF OF THEOREM F.1

*Proof.* The proof of Theorem F.1 relies on the recent usage of the BOB framework that was firstly introduced in Cheung et al. (2019) and then widely used in various bandit-based model selection work Ding et al. (2022a); Zhao et al. (2020). To be consistent we would use the notations in Algorithm 3 in this proof, and we would also recall these notations here for readers' convenience: for the $i$-th bottom epoch, we assume the candidate $H_{j_i}$ is pulled from the set $J$ in the beginning, where $j_i$ is the index of the pulled candidate. At round $t$, given the current bottom epoch length $H_{j_i}$ for some $i$, we pull the arm $v_t(H_{j_i}) \in A$ and then collect the stochastic reward $Y_t$. We also define $c_i(T)$ as the number of change points during each top epoch, and hence it naturally holds that $\sum_{i=1}^{\lceil T/H_0 \rceil} c_i(T) = c(T)$. Given these notations, the expected cumulative regret could be decomposed

into the following two parts:

$$R_L(T) = \mathbb{E}\left[\sum_{t=1}^{T} f_t(v_t^*) - f_t(v_t)\right] = \mathbb{E}\left[\sum_{i=1}^{\lceil T/H_0 \rceil} \sum_{t=(i-1)H_0+1}^{\min\{T,iH_0\}} f_t(v_t^*) - f_t(v_t(H_{j_i}))\right]$$

$$= \underbrace{\mathbb{E}\left[\sum_{i=1}^{\lceil T/H_0 \rceil} \sum_{t=(i-1)H_0+1}^{\min\{T,iH_0\}} f_t(v_t^*) - f_t(v_t(H^*))\right]}_{\text{Quantity (I)}}$$

$$+ \underbrace{\mathbb{E}\left[\sum_{i=1}^{\lceil T/H_0 \rceil} \sum_{t=(i-1)H_0+1}^{\min\{T,iH_0\}} f_t(v_t(H^*)) - f_t(v_t(H_{j_i}))\right]}_{\text{Quantity (II)}}, \tag{16}$$

where $H^*$ could be any restarting period in $J$, and we expect it could approximate the optimal choice $H^{\text{opt}} = (T/c(T))^{(p_u+2)/(p_u+3)}$ in Theorem 4.1. (Here we replace $p_{z,*}$ by $p_u$ in Theorem 4.1 since the underlying $p_{z,*}$ is mostly unspecified in reality.) According to the proof of Theorem 4.1 in Appendix G, the Quantity (I) could be bounded as:

$$\mathbb{E}\left[\sum_{i=1}^{\lceil T/H_0 \rceil} \sum_{t=(i-1)H_0+1}^{\min\{T,iH_0\}} f_t(v_t^*) - f_t(v_t(H^*))\right] \leq \sum_{i=1}^{\lceil T/H_0 \rceil} H^* c_i(T) + \frac{H_0}{H^*}(H^*)^{\frac{p_u+2}{p_u+4}}$$

$$= H^* c(T) + T(H^*)^{-\frac{1}{p_u+2}}$$

However, it is clear that each candidate in $J$ could at most be the length of top epoch size $H_0$, which we set to be $\lceil T^{(p_u+2)/(p_u+4)} \rceil$, and hence it would be more challenging if the optimal choice $H^{\text{opt}} = (T/c(T))^{(p_u+2)/(p_u+3)}$ is larger than $H_0$. To deal with this issue, we bound the expected cumulative regret in two different cases separately:

(1) If $H^{\text{opt}} = (T/c(T))^{(p_u+2)/(p_u+3)} \leq H_0$, which is equivalent to

$$\left(\frac{T}{c(T)}\right)^{\frac{p_u+2}{p_u+3}} \leq H_0 \Leftrightarrow \left(\frac{T}{c(T)}\right)^{\frac{p_u+2}{p_u+3}} \leq T^{\frac{p_u+2}{p_u+4}} \Leftrightarrow c(T) \geq T^{\frac{1}{p_u+4}},$$

then we know that there exists some $H^+ \in J$ such that $H^+ \leq (T/c(T))^{(p_u+2)/(p_u+3)} \leq 2H^+$. By setting $H^* = H^+$, the Quantity (I) could be bounded as:

$$\text{Quantity (I)} = \tilde{O}\left(H^+ c(T) + T(H^*)^{-\frac{1}{p_u+2}}\right)$$

$$= \tilde{O}\left(H^{\text{opt}} c(T) + T(H^{\text{opt}})^{-\frac{1}{p_u+2}}\right) = \tilde{O}\left(T^{\frac{p_u+2}{p_u+3}} c(T)^{\frac{1}{p_u+3}}\right).$$

For the Quantity (II), we could bound it based on the results in Auer et al. (2002b). Specifically, from Corollary 3.2 in Auer et al. (2002b), the expected cumulative regret of EXP3 could be upper bounded by $2Q\sqrt{(e-1)LK\ln(K)}$, where $Q$ is the maximum absolute sum of rewards in any epoch, $L$ is the number of rounds and $K$ is the number of arms. Under our setting, we can set $Q = H_0$, $L = \lceil T/H_0 \rceil$ and $K = |J| = O(\ln(H_0))$. So we could bound Quantity (II) as:

$$\mathbb{E}\left[\sum_{i=1}^{\lceil T/H_0 \rceil} \sum_{t=(i-1)H_0+1}^{\min\{T,iH_0\}} f_t(v_t(H^*)) - f_t(v_t(H_{j_i}))\right] \leq 2\sqrt{e-1}H_0\sqrt{\frac{T}{H_0}|J|\ln(|J|)} = \tilde{O}(\sqrt{TH_0})$$

$$= \tilde{O}\left(T^{\frac{p_u+3}{p_u+4}}\right) = \tilde{O}\left(T^{\frac{p_u+2}{p_u+3}} T^{\frac{1}{(p_u+3)(p_u+4)}}\right) = \tilde{O}\left(T^{\frac{p_u+2}{p_u+3}} c(T)^{\frac{1}{p_u+3}}\right), \tag{17}$$

where we have the last equality since we assume that $c(T) \geq T^{1/(p_u+4)}$. Therefore, we have finished the proof for this case. (2) If $H^{\text{opt}} = (T/c(T))^{(p_u+2)/(p_u+3)} > H_0$, which is equivalent to

---

**Algorithm 3** Zooming TS algorithm with Double Restarts

---

**Input:** Time horizon $T$, space $A$, upper bound $p_u \le p_c$.

**Initialization:** the (top) epoch size $H_0 = \lceil T^{(p_u+2)/(p_u+4)} \rceil$, $N = \lceil \log_2(H_0) \rceil + 1$, $J = \{H_i = \lceil H_0/2^{i-1} \rceil\}_{i=1}^N$.

1: Initialize the exponential weights $w_j(1) = 1$ for $j = 1, \ldots, |J|$.

2: Initialize the exploration parameter for the EXP3 algorithm as $\alpha = \min\left\{1, \sqrt{\frac{|J| \log(|J|)}{(e-1)\lceil T/H_0 \rceil}}\right\}$.

3: **for** $i = 1$ **to** $\lceil T/H_0 \rceil$ **do**

4:     Update probability distribution for selecting candidates in $J$ based on EXP3 as:

$$p_j(i) = \frac{\alpha}{|J|} + (1-\alpha)\frac{w_j(i)}{\sum_{k=1}^{|J|} w_k(i)}, \; j = 1, \ldots, |J|.$$

5:     Pull $j_i$ from $\{1, 2, \ldots, |J|\}$ according to the probability distribution $\{p_j(i)\}_{j=1}^{|J|}$.

6:     Run Zooming TS algorithm with Restarts using the (bottom) epoch size $H_{j_i}$ for $t = (i-1)H_0 + 1$ **to** $\min\{T, iH_0\}$, and collect the pulled arm $v_t(H_{j_i})$ and reward $Y_t$ at each iteration.

7:     Update components in EXP3: $r_j(i) = 0$ for all $j \ne j_i$; $r_j(i) = \sum_{k=(i-1)H_0+1}^{\min\{T, iH_0\}} Y_k/p_j(i)$ if $j = j_i$, and then

$$w_j(i+1) = w_j(i) \exp\left(\frac{\alpha}{|J|} r_j(i)\right), \; j = 1, \ldots, |J|.$$

8: **end for**

---

$$\left(\frac{T}{c(T)}\right)^{\frac{p_u+2}{p_u+3}} > H_0 \Leftrightarrow \left(\frac{T}{c(T)}\right)^{\frac{p_u+2}{p_u+3}} > T^{\frac{p_u+2}{p_u+4}} \Leftrightarrow c(T) < T^{\frac{1}{p_u+4}},$$

then we know that $H^{\text{opt}}$ is greater than all candidates in $J$, which means that we could not bound the Quantity (I) based on the previous argument. By simply using $H^* = H_0$, it holds that

$$\text{Quantity (I)} = \tilde{O}\left(H_0 c(T) + T \cdot H_0^{-\frac{1}{p_u+2}}\right) = \tilde{O}\left(T^{\frac{p_u+3}{p_u+4}}\right).$$

For Quantity (II), based on Eqn. (17), we have

$$\text{Quantity (II)} = \tilde{O}\left(T^{\frac{p_u+3}{p_u+4}}\right).$$

Combining the case (1) and (2), it holds that

$$R_L(T) \le \begin{cases} T^{\frac{p_u+2}{p_u+3}} c(T)^{\frac{1}{p_u+3}}, & c(T) \ge T^{\frac{1}{p_u+4}}, \\ T^{\frac{p_u+3}{p_u+4}}, & c(T) < T^{\frac{1}{p_u+4}}. \end{cases}$$

And this concludes our proof. $\qquad\qquad\qquad\qquad\qquad\qquad\qquad\qquad\qquad\qquad\qquad\qquad$ $\square$

## G ANALYSIS OF THEOREM 4.2

### G.1 ADDITIONAL LEMMA

**Lemma G.1** (Proposition 1 in Li et al. (2017)). *Define $V_{n+1} = \sum_{t=1}^n X_t X_t^T$, where $X_t$ is drawn IID from some distribution in unit ball $\mathbb{B}^d$. Furthermore, let $\Sigma := E[X_t X_t^T]$ be the second moment matrix, let $B, \delta_2 > 0$ be two positive constants. Then there exists positive, universal constants $C_1$ and $C_2$ such that $\lambda_{\min}(V_{n+1}) \ge B$ with probability at least $1 - \delta_2$, as long as*

$$n \ge \left(\frac{C_1\sqrt{d} + C_2\sqrt{\log(1/\delta_2)}}{\lambda_{\min}(\Sigma)}\right)^2 + \frac{2B}{\lambda_{\min}(\Sigma)}.$$

**Lemma G.2** (Theorem 2 in Abbasi-Yadkori et al. (2011)). *For any $\delta < 1$, under our problem setting in Section 3, it holds that for all $t > 0$,*

$$\left\| \hat{\theta}_t - \theta^* \right\|_{V_t} \leq \beta_t(\delta),$$

$$\forall x \in \mathbb{R}^d, |x^\top (\hat{\theta}_t - \theta^*)| \leq \|x\|_{v_t^{-1}} \beta_t(\delta),$$

*with probability at least $1 - \delta$, where*

$$\beta_t(\delta) = \sigma \sqrt{\log \left( \frac{(\lambda + t)^d}{\delta^2 \lambda^d} \right)} + \sqrt{\lambda} S.$$

In this subsection we denote $\alpha^*(\delta) := \beta_T(\delta)$.

**Lemma G.3** (Filippi et al. (2010)). *Let $\lambda > 0$, and $\{x_i\}_{i=1}^t$ be a sequence in $\mathbb{R}^d$ with $\|x_i\| \leq 1$, then we have*

$$\sum_{s=1}^t \|x_s\|_{V_s^{-1}}^2 \leq 2 \log \left( \frac{\det(V_{t+1})}{\det(\lambda I)} \right) \leq 2d \log \left( 1 + \frac{t}{\lambda} \right),$$

$$\sum_{s=1}^t \|x_s\|_{V_s^{-1}} \leq \sqrt{T \left( \sum_{s=1}^t \|x_s\|_{V_s^{-1}}^2 \right)} \leq \sqrt{2dt \log \left( 1 + \frac{t}{\lambda} \right)}.$$

**Lemma G.4** (Agrawal & Goyal (2013)). *For a Gaussian random variable $Z$ with mean $m$ and variance $\sigma^2$, for any $z \geq 1$,*

$$P(|Z - m| \geq z\sigma) \leq \frac{1}{\sqrt{\pi} z} e^{-z^2/2}.$$

**Lemma G.5** (Adapted from Lemma G.2). *For any $\delta < 1$, under our problem setting in Section 3 with the regularization hyper-parameter $\lambda \in [\lambda_{\min}, \lambda_{\max}] (\lambda_{\min} > 0)$, it holds that for all $t > 0$,*

$$\left\| \hat{\theta}_t - \theta^* \right\|_{V_t} \leq \beta_t(\delta),$$

$$\forall x \in \mathbb{R}^d, |x^\top (\hat{\theta}_t - \theta^*)| \leq \|x\|_{V_t^{-1}} \beta_t(\delta),$$

*with probability at least $1 - \delta$, where*

$$\beta_t(\delta) = \sigma \sqrt{\log \left( \frac{(\lambda_{\min} + t)^d}{\delta^2 \lambda_{\min}^d} \right)} + \sqrt{\lambda_{\max}} S.$$

*Proof.* The proof of this Lemma is trivial given Lemma G.2. For any $\lambda \in [\lambda_{\min}, \lambda_{\max}]$, according to Lemma G.2 it holds that, for all $t > 0$,

$$\left\| \hat{\theta}_t - \theta^* \right\|_{V_t} \leq \beta_t(\delta),$$

$$\forall x \in \mathbb{R}^d, |x^\top (\hat{\theta}_t - \theta^*)| \leq \|x\|_{V_t^{-1}} \beta_t(\delta),$$

with probability at least $1 - \delta$, where

$$\beta_t(\delta) = \sigma \sqrt{\log \left( \frac{(\lambda + t)^d}{\delta^2 \lambda^d} \right)} + \sqrt{\lambda} S \leq \sigma \sqrt{\log \left( \frac{(\lambda_{\min} + t)^d}{\delta^2 \lambda_{\min}^d} \right)} + \sqrt{\lambda_{\max}} S.$$

$\square$

## G.2    PROOF OF THEOREM 4.2

Recall the partition of the cumulative regret as:

$$
R(T) = \underbrace{\mathbb{E}\left[\sum_{t=1}^{T_1} \left(\mu(x_{t,*}^\top \theta^*) - \mu(x_t{}^\top \theta^*)\right)\right]}_{\text{Quantity (A)}} + \underbrace{\mathbb{E}\left[\sum_{t=T_1+1}^{T} \left(\mu(x_{t,*}^\top \theta^*) - \mu(x_t(\alpha^*(t)|\mathcal{F}_t^*)^\top \theta^*)\right)\right]}_{\text{Quantity (B)}}
$$

$$
+ \underbrace{\mathbb{E}\left[\sum_{t=T_1+1}^{T} \left(\mu\left(x_t(\alpha^*(t)|\mathcal{F}_t^*)^\top \theta^*\right) - \mu(x_t(\alpha^*(t)|\mathcal{F}_t)^\top \theta^*)\right)\right]}_{\text{Quantity (C)}}
$$

$$
+ \underbrace{\mathbb{E}\left[\sum_{t=T_1+1}^{T} \left(\mu\left(x_t(\alpha^*(t)|\mathcal{F}_t)^\top \theta^*\right) - \mu(x_t(\alpha(i_t)|\mathcal{F}_t)^\top \theta^*)\right)\right]}_{\text{Quantity (D)}}.
$$

For Quantity (A), it could be easily bounded by the length of warming up period as:

$$
\mathbb{E}\left[\sum_{t=1}^{T_1} \left(\mu(x_{t,*}^\top \theta^*) - \mu(x_t{}^\top \theta^*)\right)\right] \le T_1 = O(T^{\frac{2}{p+3}}) \le O(T^{\frac{p+2}{p+3}}). \tag{18}
$$

For Quantity (B), it depicts the cumulative regret of the contextual bandit algorithm that runs with the theoretical optimal hyperparameter $\alpha^*(t)$ all the time. Therefore, if we implement any state-of-the-arm contextual generalized linear bandit algorithms (e.g. Filippi et al. (2010); Li et al. (2010; 2017)), it holds that

$$
\mathbb{E}\left[\sum_{t=T_1+1}^{T} \left(\mu(x_{t,*}^\top \theta^*) - \mu(x_t(\alpha^*(t)|\mathcal{F}_t^*)^\top \theta^*)\right)\right] \le \tilde{O}(\sqrt{T - T_1}) = \tilde{O}(\sqrt{T}). \tag{19}
$$

For Quantity (C), it represents the cumulative difference of regret under the theoretical optimal hyperparameter combination $\alpha^*(t)$ with two lines of history $\mathcal{F}$ and $\mathcal{F}^*$. Note for most GLB algorithms, the most significant hyperparameter is the exploration rate, which directly affect the decision-making process. Regarding the regularization hyperparameter $\lambda$, it is used to make $V_t$ invertible and hence would be set to 1 in practice. And in the long run it would not be influential. Moreover, there is commonly no theoretical optimal value for $\lambda$, and it could be set to an arbitrary constant in order to obtain the $\tilde{O}(\sqrt{T})$ bound of regret. For theoretical proof, this hyperparameter ($\lambda$) is also not significant: for example, if the search interval for $\lambda$ is $[\lambda_{\min}, \lambda_{\max}]$, then we can easily modify the Lemma G.3 as:

$$
\sum_{s=1}^{t} \|x_s\|_{V_s^{-1}}^2 \le 2\log\left(\frac{\det(V_{t+1})}{\det(\lambda I)}\right) \le 2d\log\left(1 + \frac{t}{\lambda_{\min}}\right),
$$

$$
\sum_{s=1}^{t} \|x_s\|_{V_s^{-1}} \le \sqrt{T\left(\sum_{s=1}^{t} \|x_s\|_{V_s^{-1}}^2\right)} \le \sqrt{2dt\log\left(1 + \frac{t}{\lambda_{\min}}\right)}.
$$

We will offer a more detailed explanation to this fact in the following proof of bounding Quantity (C). Furthermore, other parameters such as the stepsize in a loop of gradient descent will not be crucial either since the final result would be similar after the convergence criterion is met. Therefore, w.l.o.g we would only assume there is only one exploration rate hyperparameter here to bound Quantity (C). Recall that $\alpha(t)$ is the combination of all hyperparameters, and hence we could denote this exploration rate hyperparameter as $\alpha(t)$ in this part since there is no more other hyperparameter. Here we would use LinUCB and LinTS for the detailed proof, and note that regret bound of all other UCB and TS algorithms could be similarly deduced. We first reload some notations: recall we denote

$V_t = \lambda I + \sum_{i=1}^{t-1} x_i x_i^\top, \hat{\theta}_t = V_t^{-1} \sum_{i=1}^{t-1} x_i y_i$ where $x_t$ is the arm we pulled at round $t$ by using our tuned hyperparameter $\alpha(i_t)$ and the history based on our framework all the time. And we denote

$$X_t = \arg\max_{x \in \mathcal{X}_t} x^\top \hat{\theta}_t + \alpha^*(t) \|x\|_{V_t^{-1}}$$

Similarly, we denote $\tilde{V}_t = \lambda I + \sum_{i=1}^{t-1} \tilde{X}_i \tilde{X}_i^\top, \tilde{\theta}_t = \tilde{V}_t^{-1} \sum_{i=1}^{t-1} \tilde{X}_i \tilde{y}_i$, where $\tilde{X}_t$ is the arm we pulled by using the theoretical optimal hyperparameter $\alpha^*(t)$ under the history of always using $\{\alpha^*(s)\}_{s=1}^{t-1}$, and $\tilde{y}_t$ is the corresponding payoff we observe at round $t$. Therefore, it holds that,

$$\tilde{X}_t = \arg\max_{x \in \mathcal{X}_t} x^\top \tilde{\theta}_t + \alpha^*(t) \|x\|_{\tilde{V}_t^{-1}} .$$

By using these new definitions, the Quantity (C) could be formulated as:

$$\mathbb{E}\left[ \underbrace{\sum_{t=T_1+1}^{T} \left( \mu\left( x_t(\alpha^*(t)|\mathcal{F}_t^*)^\top \theta^* \right) - \mu(x_t(\alpha^*(t)|\mathcal{F}_t)^\top \theta^*) \right)}_{\text{Quantity (C)}} \right] = \mathbb{E}\left[ \sum_{t=T_1+1}^{T} \mu(\tilde{X}_t^\top \theta^*) - \mu(X_t^\top \theta^*) \right]$$

For LinUCB, since the Lemma G.2 holds for any sequence $(x_1, \ldots, x_t)$, and hence we have that with probability at least $1 - \delta$,

$$\left\| \hat{\theta} - \theta \right\|_{V_t} \le \beta_t(\delta) \le \alpha^*(T, \delta), \tag{20}$$

where

$$\beta_t(\delta) = \sigma \sqrt{\log\left( \frac{(\lambda + t)^d}{\delta^2 \lambda^d} \right)} + \sqrt{\lambda} S = \alpha^*(t).$$

And we will omit $\delta$ for simplicity. For LinUCB, we have that

$$X_t^\top \hat{\theta}_t + \alpha^*(t) \|X_t\|_{V_t^{-1}} \ge \tilde{X}_t^\top \hat{\theta}_t + \alpha^*(t) \left\| \tilde{X}_t \right\|_{V_t^{-1}}$$
$$\ge \tilde{X}_t^\top \theta^* + \alpha^*(t) \left\| \tilde{X}_t \right\|_{V_t^{-1}} + \tilde{X}_t^\top (\hat{\theta}_t - \theta^*) \ge \tilde{X}_t^\top \theta^*.$$

Therefore, it holds that

$$X_t^\top \theta^* + \alpha^*(t) \|X_t\|_{V_t^{-1}} + X_t^\top (\hat{\theta}_t - \theta^*) \ge \tilde{X}_t^\top \theta^*$$
$$X_t^\top \theta^* + 2\alpha^*(t) \|X_t\|_{V_t^{-1}} \ge \tilde{X}_t^\top \theta^*,$$

which implies that

$$(\tilde{X}_t - X_t)^\top \theta^* \le 2\alpha^*(T) \|X_t\|_{V_t^{-1}} .$$

By Lemma G.3 and choosing $T_1 = T^{2/(p+3)}$, it holds that,

$$\sum_{t=T_1+1}^{T} \|X_t\|_{V_t^{-1}} \le \sum_{t=T_1+1}^{T} \|X_t\| \sqrt{\lambda_{\min}(V_t)} = O(T \times T^{-1/(p+3)}) = O(T^{(p+2)/(p+3)}).$$

And then it holds that,

$$\sum_{t=T_1+1}^{T} \left( \tilde{X}_t^T \theta - X_t \theta \right) = \tilde{O}\left( \alpha^*(T) \sum_{t=T_1+1}^{T} \left\| \tilde{X}_t \right\|_{V_t^{-1}} \right) = \tilde{O}(T^{(p+2)/(p+3)}). \tag{21}$$

Note $\beta_t(\delta)$ contain the regularizer parameter $\lambda$, and it's often set to some constant (e.g. 1) in practice. If we tune $\lambda$ in the search interval $[\lambda_{\min}, \lambda_{\max}]$, then we can still have the identical bound as in Eqn. (20) by using the fact that

$$\beta_t(\delta) = \sigma \sqrt{\log\left( \frac{(\lambda + t)^d}{\delta^2 \lambda^d} \right)} + \sqrt{\lambda} S \le \sigma \sqrt{\log\left( \frac{(\lambda_{\min} + t)^d}{\delta^2 \lambda_{\min}^d} \right)} + \sqrt{\lambda_{\max}} S.$$

This result is deduced in our Lemma G.5, which implies that tuning the regularizer hyperparameter would not affect the order of final regret bound in Eqn. (21). Therefore, as we mentioned earlier, we could only consider the exploration rate as the unique hyperparameter for theoretical analysis.

For LinTS, we have that

$$
\begin{aligned}
X_t^\top \hat{\theta}_t + \alpha^*(T) \left\| X_t \right\|_{V_t^{-1}} Z_t &\geq \tilde{X}_t^\top \hat{\theta}_t + \alpha^*(T) \left\| \tilde{X}_t \right\|_{V_t^{-1}} \tilde{Z}_t \\
&\geq \tilde{X}_t^\top \theta^* + \alpha^*(T) \left\| \tilde{X}_t \right\|_{V_t^{-1}} \tilde{Z}_t + \tilde{X}_t^\top (\hat{\theta}_t - \theta^*) \\
&\geq \tilde{X}_t^\top \theta^* + \alpha^*(T) \left\| \tilde{X}_t \right\|_{V_t^{-1}} \tilde{Z}_t + \left\| \tilde{X}_t \right\|_{V_t^{-1}} \left\| \hat{\theta}_t - \theta^* \right\|_{V_t} \\
&\geq \tilde{X}_t^\top \theta + (\alpha^*(T) \tilde{Z}_t - \alpha^*(T)) \left\| \tilde{X}_t \right\|_{V_t^{-1}},
\end{aligned}
$$

where $Z_t$ and $Z_{t,*}$ are IID normal random variables, $\forall t$. And then we could deduce that

$$
X_t^\top \theta^* + \alpha^*(T) \left\| X_t \right\|_{V_t^{-1}} Z_t + X_t^\top (\hat{\theta}_t - \theta^*) \geq \tilde{X}_t^\top \theta + (\alpha^*(T) \tilde{Z}_t - \alpha^*(T)) \left\| \tilde{X}_t \right\|_{V_t^{-1}}
$$

$$
X_t^\top \theta^* + \alpha^*(T) \left\| X_t \right\|_{V_t^{-1}} Z_t + \alpha^*(T) \left\| X_t \right\|_{V_t^{-1}} \geq \tilde{X}_t^\top \theta + (\alpha^*(T) \tilde{Z}_t - \alpha^*(T)) \left\| \tilde{X}_t \right\|_{V_t^{-1}}
$$

$$
(\tilde{X}_t - X_t)^\top \theta^* \leq (\alpha^*(T) - \alpha^*(T) \tilde{Z}_t) \left\| \tilde{X}_t \right\|_{V_t^{-1}} + (\alpha^*(T) + \alpha^*(T) Z_t) \left\| X_t \right\|_{V_t^{-1}} := K_t
$$

where $K_t$ is normal random variable with

$$
\mathbb{E}(K_t) \leq 2\alpha(T) T^{-1/(p+3)}, \ \mathrm{SD}(K_t) \leq \sqrt{2}\alpha^* T^{-1/(p+3)}.
$$

Consequently, we have

$$
\sum_{t=T_1+1}^{T} \left( \tilde{X}_t^T \theta - X_t^T \theta \right) \leq \sum_{t=T_1+1}^{T} K_t := K
$$

$$
\mathbb{E}(K) = 2\alpha^*(T) T^{(p+2)/(p+3)} = \tilde{O}(T^{\frac{p+2}{p+3}}), \ \mathrm{SD}(K) \leq \sqrt{2}\alpha^* T^{\frac{p+1}{2p+6}} = O(T^{\frac{p+1}{2p+6}}).
$$

Based on Lemma G.4, we have

$$
P\left( \sum_{t=T_1+1}^{T} \left( \tilde{X}_t^T \theta - X_t^T \theta \right) \geq K > (2\alpha^* + \sqrt{2}) T^{\frac{p+2}{p+3}} \right) \leq \frac{1}{c\sqrt{\pi}\sqrt{T}} e^{-c^2 T/2}. \tag{22}
$$

This probability upper bound is minimal and negligible, which means the bound on its expected value (Quantity (C)) could be easily deduced. Note we could use this procedure to bound the regret for other UCB and TS bandit algorithms, since most of the proof for GLB algorithms are closely related to the rate of $\sum_{t=T_1+1}^{T} \left\| X_t \right\|_{V_t^{-1}}$ and the consistency of $\hat{\theta}_t$. In conclusion, we have that Quantity (C) could be upper bounded by the order $\tilde{O}(T^{\frac{p+2}{p+3}})$.

For Quantity (D), which is the extra regret we paid for hyperparameter tuning in theory. Recall we assume $\mu(x_t(\alpha | \mathcal{F}_t)^\top \theta^*) = g_t(\alpha) + \eta_{\mathcal{F}_t, \alpha}$ for some time-dependent Lipschitz function $g_t$. And $(\eta_{\mathcal{F}_t, \alpha} - \mathbb{E}[\eta_{\mathcal{F}_t, \alpha}])$ is IID sub-Gaussian with parameter $\tau^2$ where $\mathbb{E}[\eta_{\mathcal{F}_t, \alpha}]$ depends on the history $\mathcal{F}_t$. Denote $\nu_{\mathcal{F}_t, \alpha} = \eta_{\mathcal{F}_t, \alpha} - \mathbb{E}[\eta_{\mathcal{F}_t, \alpha}]$ is the IID sub-Gaussian random variable with parameter $\tau^2$, then we have that

$$
y_t = g_t(\alpha(i_t)) + \nu_{\mathcal{F}_t, \alpha(i_t)} + E[\eta_{\mathcal{F}_t, \alpha(i_t)}] + \epsilon_t
$$

Since $\nu_{\mathcal{F}_t, \alpha(i_t)}, \epsilon_t$ is IID sub-Gaussian random variable independent with $\mathcal{F}_t$, we denote $\tilde{\epsilon}_{\mathcal{F}_t, \alpha(i_t)} = \nu_{\mathcal{F}_t, \alpha(i_t)} + \epsilon_t$ as the IID sub-Gaussian noise with parameter $\tau^2 + \sigma^2$. And then we have

$$
y_t = g_t(\alpha(i_t)) + E[\eta_{\mathcal{F}_t, \alpha(i_t)}] + \tilde{\epsilon}_{\mathcal{F}_t, \alpha(i_t)}, \quad \mathbb{E}(y_t) = g_t(\alpha(i_t)) + E[\eta_{\mathcal{F}_t, \alpha(i_t)}]
$$

$$
\mu(x_t(\alpha | \mathcal{F}_t)^\top \theta^*) = g_t(\alpha) + E[\eta_{\mathcal{F}_t, \alpha}].
$$

For Quantity (D), recall it could be formulated as:

$$\underbrace{\mathbb{E}\left[\sum_{t=T_1+1}^{T}\left(\mu\left(x_t(\alpha^*(t)|\mathcal{F}_t)^\top\theta^*\right)-\mu(x_t(\alpha(i_t)|\mathcal{F}_t)^\top\theta^*))\right)\right]}_{\text{Quantity (D)}}.$$

Since both terms in Quantity (D) are based on the same line of history $\mathcal{F}_t$ at iteration $t$, and the value of $E[\eta_{\mathcal{F}_t,\alpha}]$ only depends on the history filtration $\mathcal{F}_t$ but not the value of $\alpha$. Therefore, it holds that

$$\underbrace{\mathbb{E}\left[\sum_{t=T_1+1}^{T}\left(\mu\left(x_t(\alpha^*(t)|\mathcal{F}_t)^\top\theta^*\right)-\mu(x_t(\alpha(i_t)|\mathcal{F}_t)^\top\theta^*))\right)\right]}_{\text{Quantity (D)}}=\sum_{t=T_1+1}^{T}g_t(\alpha^*(t))-\mathbb{E}[g_t(\alpha(i_t))]$$
$$\leq\sum_{t=T_1+1}^{T}\sup_{\alpha\in A}g_t(\alpha)-\mathbb{E}[g_t(\alpha(i_t))].$$

Therefore, Quantity (D) could be regarded as the cumulative regret of a non-stationary Lipschitz bandit and the noise is IID sub-Gaussian with parameter $\tau_0^2=(\tau^2+\sigma^2)$. We assume that, under the *switching* environment, the Lipschitz function $g_t(\cdot)$ would be piecewise stationary and the number of change points is of scale $\tilde{O}(1)$. Therefore, Quantity (D) can be upper bounded the cumulative regret of our Zooming TS algorithm with restarted strategy given $c(T)=\tilde{O}(1)$. By choosing $T_2=(T-T_1)^{(p+2)/(p+3)}=\Theta(T^{(p+2)/(p+3)})$, and according to Theorem 4.1, it holds that,

$$\sum_{t=T_1+1}^{T}\sup_{\alpha\in A}g_t(\alpha)-\mathbb{E}[g_t(\alpha(i_t))]\leq\tilde{O}\left(T^{\frac{p+2}{p+3}}\right). \tag{23}$$

By combining the results deduced in Eqn. (18), Eqn. (19), Eqn. (21) (or Eqn. (22)) and Eqn. (23), we finish the proof of Theorem 4.2 for linear bandits. For generalized linear bandits, under the default and standard assumption in the generalized linear bandit literature that the derivative of $\mu(\cdot)$ could be upper bounded by some constant given $|x|\leq S$, the regret could be bounded by further multiplying a constant in the same order.

$\square$