# OpenReview forum: "Dynamic Continuous Hyperparameter Tuning for Generalized Linear Contextual Bandits"
_ICLR.cc/2024/Conference — Submitted to ICLR 2024_

### Official Review · Reviewer_m7nW · 2023-10-27

**Soundness:** 3 good
**Presentation:** 3 good
**Contribution:** 3 good
**Rating:** 5
**Confidence:** 4

**Summary:**

In response to some hyperparameter selection issues in bandit problems, this paper notes that theoretically derived parameter often yield unsatisfactory results in practice. At the same time, existing methods for hyperparameter search often lack theoretical guarantees and cannot be directly applied to real-world problems. To address this, the paper introduces a a double-layer BOB framework named CDT, for learning the hyperparameters. This models the hyperparameter selection issue as a continuum-arm Lipschitz bandit problem, ultimately providing sub-linear regret theoretical guarantees. Moreover, the method demonstrates significant performance improvements in experiments.

**Strengths:**

The paper introduces an algorithm for tuning parameters in a continuous parameter space, provides the corresponding theoretical guarantees, and demonstrates performance improvement through experiments.

**Weaknesses:**

It seems that your meta-algorithm's parameter $H$ still requires some prior knowledge about $T$ and $p_{z,*}$, which doesn’t fully resolve the issue of parameter tuning. In practical scenarios, it is impossible to know the total iteration number $T$ and other prior knowledge in advance. Addressing this issue would need an additional layer of BOB, resulting in a three-layer algorithm structure. Are you sure that such an algorithm can run efficiently and can still have a good performance? Moreover, this approach may impose certain constraints on the total duration $T$, such as requiring $T$ to be larger than certain values.

**Questions:**

1. The theoretical results of the CDT algorithm only provide sublinear outcomes, and it seems like it can't recover the optimal results of the base algorithms. However, your experiments indicate that CDT is optimal, which feels somewhat hard to explain.

2. The author mentions in the contributions section that the algorithm presented in the paper is efficient. However, there is no comparison of running times in the experiments. This is quite crucial for bandit settings. Therefore, I would suggest that the author add a comparison of the algorithm's running time in the experiments.

3. There is a typo in Notation: write \top as T

---

> ### Author Response · Authors · 2023-11-18
> **Thank you very much for your valuable comments. Please see our responses to your feedbacks**
>
> Thank you for your insightful comments on our work, and we are pleased to know that you find our work solves an important practical problem and achieves significant performance improvements in comprehensive experiments. Please see our responses to your concerns as follows:
>
>
> Weakness:
>
>
> We appreciate your careful review. To use our algorithm 1 as the meta-algorithm in our CDT framework (algorithm 2), the values of $T_1$ (warming time) and $T_2$ (epoch length) only depend on the number of hyperparameters we want to tune ($p$), which is known in the beginning. This is because we can safely use an upper bound $p$ of the unknown zooming dimension $p_{z,\*}$ in practice, and the final regret deduced in our Theorem 4.2 will depend on $p$ instead of $p_{z,*}$. For the issue of unknown $T$, we can utilize the widely-used doubling trick [1] when $T$ is unknown with the same regret bound held. We will mention this trick in the revision of our work.
>
>
> In section 4.1, we mention that the choice of $H$ could be handled by introducing an extra layer for the completeness of theory, and we provide a detailed analysis in Appendix F with solid theoretical support. However, we don’t use this procedure in our CDT framework due to concerns about the efficiency of the three-layer structure in practice as well. As in Section 4.1, we illustrate that the choice of $H$ could be handled by introducing this extra layer **mostly for the completeness of theoretical analysis**.
>
>
> For the restriction on the value of $T$, our algorithm doesn’t have to assume the value of $T$ is larger than certain values in theory, and the warming-up time $T_1$ is always strictly less than $T$. For practice, it is noteworthy that bandit algorithms cannot achieve good performance if $T$ is small since they estimate the model parameters on the fly, not to mention the more difficult model selection task for bandits. From Figure 1, we can clearly observe that the cumulative regrets of our CDT framework are smaller after some warm-up iterations in most cases, which validates that our CDT could perform better than all existing baselines regardless of the magnitude of $T$.
>
> [1]. The adversarial multi-armed bandit problem. P. Auer et al., IEEE Annual Symposium on Foundations of Computer Science, 1995.
>
>
> Q1:
>
>
> Thank you for your insightful questions. It is widely known that there is a huge discrepancy between the theoretical analysis and practical performance for bandit algorithms. From Figure 1, we can see that the theoretically derived hyperparameters usually yield terrible or even linear cumulative regret for most algorithms while they achieve the optimal regret bound in theory, and this is because most theoretical optimal settings are way too conservative in practice. To obtain better practical performance, intuitively we should choose less conservative exploration hyperparameters, which will lead to worse regret bounds. This fact is the motivation why we study this intriguing practical problem: how to choose hyperparameters for bandit algorithms in practice to adapt to different environments with superior performance while some decent regret bounds can still hold. As we mention in our work, we mostly concentrate on the practical performances of our CDT framework in our paper. And according to our section 4.2 and our proof in Appendix G, it is very difficult and challenging to present the novel theoretical analysis of Theorem 4.2. Furthermore, the existing work on bandit hyperparameter tuning (e.g. "Syndicated bandits: A framework for auto-tuning hyper-parameters in contextual bandit algorithms") intrinsically can't achieve the best regret bound of the base algorithms without stringent assumptions. Specifically, Syndicated can only achieve the $\tilde O(T^{2/3})$ bound of regret with finite number of candidate values for each hyperparameter, and their regret bound will becomes infinitely large under our problem setting with continuous candidate sets. These facts also demonstrate the significance of our theoretical analysis, even though the focus of our work is mainly on practical performances.

---

> ### Author Response · Authors · 2023-11-18
> **Responses to your feedbacks Part 2**
>
> Q2
>
> **In our work we put the running time of our method with all baselines in Appendix A.4.5** for the 10 cases corresponding to Figure 1 plots 1-10. For ease of your review, we copied the results from Appendix A.4.5 in the following tables. Furthermore, we reproduced the last 4 cases in Figure 1 (plots 11-14) these days and displayed the results for completeness:
>
> LinUCB
>
> |        | Simulation | Movielens |
> |--------|------------|-----------|
> | Theory | 2.11s      | 2.17s     |
> | TL    | 2.61s      | 2.67s     |
> | OP     | 2.60s      | 2.65s     |
> | CDT    | 3.49s      | 3.39s     |
> LinTS
>
> |        | Simulation | Movielens |
> |--------|------------|-----------|
> | Theory | 2.19s      | 2.04s     |
> | TL    | 2.67s      | 2.49s     |
> | OP     | 2.62s      | 2.45s     |
> | CDT     | 3.53s      | 3.46s     |
>
> UCB-GLM
>
> |        | Simulation | Movielens |
> |--------|------------|-----------|
> | Theory | 8.34s      | 7.89s     |
> | TL    | 8.84s      | 8.44s     |
> | OP     | 8.71s      | 8.35s     |
> | CDT     | 9.57s      | 9.47s     |
>
> GLM-TSL
>
> |        | Simulation | Movielens |
> |--------|------------|-----------|
> | Theory | 305.28s      | 295.17s     |
> | TL    | 305.98s      | 295.87s     |
> | OP     | 306.02s      | 295.83s     |
> | CDT     | 307.24s      | 296.72s     |
>
> Laplace-TS
>
> |        | Simulation | Movielens |
> |--------|------------|-----------|
> | Theory | 525.62s      | 502.89s     |
> | TL    | 526.31s      | 503.45s     |
> | OP     | 526.24s      | 503.51s     |
> | CDT     | 527.15s      | 504.45s     |
>
> GLOC
>
> |        | Simulation | Movielens |
> |--------|------------|-----------|
> | Theory | 486.58s      | 474.87s     |
> | TL    | 488.31s      | 476.45s     |
> | OP     | 488.24s      | 475.51s     |
> | CDT     | 490.15s      | 478.45s     |
>
> SGD-TS
>
> |        | Simulation | Movielens |
> |--------|------------|-----------|
> | Theory | 67.42s      | 62.09s     |
> | TL    | 69.31s      | 63.85s     |
> | OP     | 69.24s      | 64.51s     |
> | CDT     | 71.15s      | 66.45s     |
>
> From the above tables, we can see that all existing methods and our CDT can run very fast in practice, and our CDT is only slightly more expensive in computation (CDT only takes few more seconds). In addition, we can observe that the main computation time comes from the contextual bandit algorithm we tune on. Specifically, GLM-TSL requires much more running time than all other methods under different tuning methods. Therefore, our CDT significantly outperforms all existing baselines without increasing much computation.
>
> Q3.
>
> Thank you very much for your careful review. We will correct this typo in the revision.
>
>
>
> We sincerely value the time you've dedicated to reviewing our work. Please let us know if our response has sufficiently resolved your concern and improved your opinion of our work, and we are more than happy to engage in any further discussion with you.

---

> > ### Author Response · Authors · 2023-11-23
> > **Thank you very much for your review. We are more than happy to receive feedbacks from you**
> >
> > Thanks a lot for dedicating your time to thoroughly review our work and for providing thoughtful comments. In response to your feedback, we have meticulously formulated comprehensive answers in our rebuttal to sufficiently address your concerns.
> >
> >
> > Specifically, we first explain the choice of $H$ in our algorithm, and point out that we introduce this extra layer (Algorithm 3) to determine $H$ mostly for the completeness of theoretical analysis. However, this is not the focus of our work, and in our main contribution CDT we don’t have to use this procedure, as we mention in the last paragraph of Section 4.1. Then we explain the discrepancy between theory and practice in bandit, and illustrate the significance and difficulty of our theoretical analysis. Finally, we point out the running time of methods used in our paper are actually reported in Appendix A.4.5, and we also showcase additional extra experimental results.
> >
> >
> > We sincerely appreciate your valuable feedbacks on our work. Since the discussion deadline is approaching, we respectfully hope you could let us know whether our response has sufficiently resolved your concerns and improved your opinion of our work. And we are more than happy to engage in any further discussions with you.

---

### Official Review · Reviewer_wfPS · 2023-11-01

**Soundness:** 3 good
**Presentation:** 3 good
**Contribution:** 3 good
**Rating:** 6
**Confidence:** 3

**Summary:**

This paper studies the problem of generalized linear bandits. Though there are optimal algorithms proved for this setting, the theoretically optimal choice of their hyper-parameters are often very conservative and useless in practice. This paper attempts to learn the hyper-parameters that are best suited for the problem instance at hand which lead to significant performance boost in comparison to theoretical choice.

**Strengths:**

- The problem they study is well motivated in practice
- Novel regret analysis is presented for handling time switching lipschitz bandits
- Extensive experiments are conducted

**Weaknesses:**

I think this work studies an important practical problem. Though I am positive in general, some clarifications regarding the current draft are needed:

- The authors propose Algorithm 1 to zoom quickly into a good area in the parameter space. However, this introduces extra hyper-parameters in Algorithm 1 which is unclear to tune. The authors say that it is possible to mitigate this issue by adding another layer of EXP3, essentially treating the epoch size candidate in Alg. 1 as arms. Then how sensitive is the overall performance wrt the parameters of outer EXP3 layer?

- A follow-up on the previous question. While the master EXP3 algorithm does compete with the base algorithms (with different epoch parameters) in terms of their actual performance during the run, this performance could be significantly worse than if the base algorithm were run on its own, updating its state after every prediction. For instance, a base algorithm which is performing bad initially but excels later on might quickly fall out of favor with the master,  never reach its good performance regime. How do you address this problem? How are the base-learners defined? Doesn't this problem persist, even if we are only trying to optimize a given bandit algorithm like linUCB over a discrete candidate set of hyper-parameters? This is essentially the same problem faced by corralling a band of bandits literature (eg. https://arxiv.org/abs/1612.06246).

- I am confused about the usefulness of having fixed length epochs between two restarts in Alg.1. For example, we can expect that after sufficient number of rounds, the expected reward is stable. So isn't it more practical to have an adaptive restart schedule where one restarts more often during earlier rounds and less toward later rounds?  Further, I agree with the authors that similar hyper-parameter choices most often can result in similar arms getting pulled. However, there could be discontinuities in the rewards as a function of hyper-parameters. So I am curious, why the authors didn't study the case where the reward at a round is also modelled as a piece-wise lipschitz function of hyper-parameters: meaning the parameter space is partitioned into several clusters. within each cluster the reward is a lipschitz function.

- How do you tune Alg.1 wrt sub-gaussian parameter $\tau$ present in the noise of Eq.(2)?

**Questions:**

see above

---

> ### Author Response · Authors · 2023-11-18
> **Thank you for your careful review. Please see our responses to your concerns**
>
> Thank you for your insightful comments. We are happy that you are positive on our contributions since you find our work is well-motivated in practice and our method performs well in extensive conducted. Please see our response to your concerns.
>
> **Weakness 1:**
>
> Thank you for your insightful questions. In Section 4.1, we illustrate that the choice of $H$ could be handled by introducing an extra layer **mostly for the completeness of theory**, and we provide a detailed analysis in Appendix F with solid theoretical support.
> We agree with the reviewer that in practice this extra outer layer may not bring many benefits since between each two outer EXP3 pulls we don’t have many iterations to explore the Lipschitz structure with restarts. However, in our CDT we don’t have to use this procedure, as we mention in the last paragraph of Section 4.1.
>
> **Weakness 2:**
> Thank you very much for your valuable comments. As we emphasize in our response to Weakness 1, we use the master EXP3 algorithm to choose the base Algorithm 1 with different epoch parameters **mostly for the completeness of our theoretical analysis**. And this is not the main concentration of our work. However, we believe the methodology of our main algorithm CDT aligns with your speculation here: instead of using the same hyperparameter combination for a certain period of time, our CDT algorithm can choose the hyperparameter values and then update its state at each iteration sequentially. This can help prevent the circumstance you mention in weakness 2 since even though an eventually good hyperparameter combination may yield bad performance in the beginning, we still have lots of iterations to update our estimate promptly, and hence this hyperparameter combination will become favorable again after rounds of updates. Moreover, the restarting method in our CDT can further help to alleviate this issue: it can discard the historical memories generated from different environments periodically so that the eventually good hyperparameter combination will not be ruled out.
>
> **Weakness 3: “have an adaptive restart schedule where one restarts more often during earlier rounds and less toward later rounds?”**
>
> Thank you for your helpful comments. We use the fixed length epochs to keep consistent with our theoretical analysis, and we can observe that it works very well in practice. To show whether it is more efficient to restart the algorithm less often in the long run, we reproduced the experiments in Figure 1 while increasing the restarting period by a multiplier of 2. The results are displayed in the following table:
>
>
> | Settings               | CDT in our paper | CDT with increasing restarting epochs   |
> |------------------------|------------|------------|
> | Simulations LinUCB     | **298.28** | 306.62     |
> | Movielens LinUCB       | 313.29 | **303.52**     |
> | Simulations LinTS      | 677.03     |  **658.71**   |
> | Movielens LinTS        | 343.18     | **341.56**     |
> | Simulations UCB-GLM    | **299.74** | 314.00     |
> | Movielens UCB-GLM      | 314.41  | **302.18**     |
> | Simulations GLM-TSL    | **339.49**     | 352.30 |
> | Movielens GLM-TSL      | **428.82**   | 433.88 |
> | Simulations Laplace-TS | 520.29     | **505.41**     |
> | Movielens Laplace-TS   | 903.16     | **900.34**     |
> | Simulations GLOC       | **415.05** | 437.94     |
> | Movielens GLOC         | 426.24     | **410.51** |
> | Simulations SGD-TS     | **414.57** | 427.62     |
> | Movielens SGD-TS       | 830.35 | **815.13**     |
>
>
>
> From the Table, we can observe that the CDT in our paper and CDT with increasing restarting epochs yield comparable performance: the CDT in our paper is better in 6 cases while the CDT with increasing restarting epochs wins 8 cases. Both algorithms can beat all the existing baselines consistently and achieve state-of-the-art results. And it is a interesting future work to study how to adaptively increase the restarting epoch with solid theoretical guarantee.
>
> Weakness 3: “There could be discontinuities in the rewards as a function of hyper-parameters.”
>
> As in "Syndicated bandits: A framework for auto-tuning hyper-parameters in contextual bandit algorithms", our work assumes that the arms in the arm set $X_t$ are drawn IID from some unknown distribution. Therefore, considering the randomness of the action set in each iteration, we can anticipate that the expected performance of the bandit algorithm should be similar with close hyperparameters given historical information (F_t) fixed. We will emphasize this point in our problem setting in the revision. We further use Figure 2 in Appendix A.2 to demonstrate the Lipschitzness of reward w.r.t hyperparameter selection empirically, and it is evident that this figure authenticates our assumption very well.

---

> ### Author Response · Authors · 2023-11-18
> **Responses to your concerns Part 2**
>
> **Weakness 4:**
>
> In practice, most studies would have the stochastic rewards bounded, e.g., in the interval (0,C) for some positive constant $C$, where we could naturally take this hyperparameter as $C/2$ when applying Algorithm 1 and Algorithm 2. Note we use this implementation in our experiments and it works very well as shown in our extensive simulations and real datasets.
>
> We really appreciate your valuable insights. Please let us know if our response has sufficiently resolved your concern and improved your opinion of our work, and we are more than happy to engage in any further discussion with you.

---

> > ### Comment · Reviewer_wfPS · 2023-11-21
> > **Reply to authors**
> >
> > Thanks for the clarifications! I think this paper takes a step in the right direction for an important practical problem. Moving forward, I would like to see if the restart idea can be theoretically backed up as an alternate solution to the problem I mentioned above regarding coraling a band of bandits applied in the context of hyper-parameter tuning.

---

> > > ### Author Response · Authors · 2023-11-22
> > > **Thank you for your response**
> > >
> > > Thank you very much for your careful review on our work, and we are happy to know that you find our work solves an important practical problem in a right direction. As we mention in Appendix B Remark B.1, the regret bound of the corralling idea depends on the worst candidates model in the tuning set, which may be unrealistic in practice. And it updates all the base models at each iteration, that would be computationally expensive. Therefore, we believe more theoretical modifications besides the restarting idea have to be made for the corralling a band of bandits under our problem setting. And we will leave this interesting topic as a future work.
> > >
> > > We appreciate your insightful comments on our work, and we are more than happy to engage in any further discussions.

---

### Official Review · Reviewer_psHy · 2023-11-02

**Soundness:** 3 good
**Presentation:** 3 good
**Contribution:** 2 fair
**Rating:** 5
**Confidence:** 3

**Summary:**

This paper explores hyperparameter tuning for generalized linear contextual bandits. The foundational concept behind the paper is the OFUL algorithm, where the chosen arm at round $t$ is:

$a_{t}=\arg\max_{a\in [K]}$ $x_{t,a}^{\top}$ $\hat \theta_{t}$ $+\alpha $ $ ||x_{t,a}||_{V{t}{-1}}$.

While setting $\alpha=\tilde{\Theta}(d)$ can yield the worst-case optimal regret bound, this parameter might result in subpar performance in real-world scenarios.

The primary objective of this paper is to adjust the parameter $\alpha$ to ensure that the algorithm maintains its theoretical guarantee while also demonstrating good empirical results. Given certain conditions, the paper achieves this goal.

**Strengths:**

See summary.

**Weaknesses:**

However, there are some notable limitations:

1. The paper overlooks key benchmarks both theoretically and experimentally:
   - Regret Bound Balancing and Elimination for Model Selection in Bandits and RL.
   - Syndicated bandits: A framework for auto-tuning hyper-parameters in contextual bandit algorithms.

2. The rationale behind the decomposition in Eq(2) suggests that “the bandit algorithm is likely to select similar arms if the hyperparameters chosen are close at round $t$.” This reasoning is not entirely convincing. When feature vectors are determined by either a non-oblivious or oblivious adversary, the claim doesn't hold up. Such an adversary can present arms where the hyperparameters are similar, yet the arms differ considerably. Hence, it's crucial for the authors to specify these conditions in the problem definition and offer a more plausible explanation for Eq(2).

3. The regret bound presented in this paper is $\Omega(T^{2/3})$. This is less favorable than the bound provided in "Syndicated bandits: A framework for auto-tuning hyper-parameters in contextual bandit algorithms," which doesn't necessitate any stringent conditions.

**Questions:**

See above.

---

> ### Author Response · Authors · 2023-11-18
> **Thank you very much for you review. Please see our response**
>
> **Weakness 1:**
>
> Thank you very much for your insightful comments. For the work “Regret Bound Balancing and Elimination for Model Selection in Bandits and RL”, we will cite it in the revision of our paper, since it discusses how to choose the confidence radius of OFUL on the fly. However, we’d like to point out that this work mainly concentrates on the general model selection in bandits and reinforcement learning, and the theoretical regret bound deduced in this work also depends on the number of candidates $M$, which is infinitely large and hence becomes meaningless in our problem setting. For experiments, the algorithm proposed in this work is also inefficient since it has to update each active base candidate at every iteration, and the implementation of this algorithm also relies on some unspecified parameters such as $\kappa$. This implies that this algorithm could not be easily adapted to the hyperparameter tuning of any contextual linear bandit. Note in Remark B.1 in Appendix B, we illustrate why the general bandit model selection method can’t be used for hyperparameter tuning of bandit algorithms, and “Syndicated bandits: A framework for auto-tuning hyper-parameters in contextual bandit algorithms” is the only existing work with theoretical guarantees that considers exactly hyperparameter tuning of bandit algorithms.
>
> For “Syndicated bandits: A framework for auto-tuning hyper-parameters in contextual bandit algorithms”. In theory, the general regret bound is of order $\tilde{O}(T^{2/3} + \sqrt{MT})$, where $M$ is the number of hyperparameters considered in the candidate set. Note that this regret bound also becomes futile in our problem setting since $M$ is infinitely large, and hence this regret bound becomes infinite. Note it is intrinsically much more difficult in theory for the infinite-arm case compared with the finite-arm case: for example, the minimax regret bound for multi-armed bandit is of order $\tilde{O}(\sqrt{T})$ while the minimax regret bound for the continuum-armed bandit in $[0,1]^d$ is of order $\tilde{O}(T^{(d+1)/(d+2)})$. This fact means that for continuum-armed bandit it is impossible to recover most of the results in multi-armed bandits in the same order. In experiments, we include the TL or Syndicated in all of our simulations and real datasets as an important baseline, and we can evidently observe the practical advantage of our proposed CDT over all existing baselines.
>
> **Weakness 2:**
>
> Thank you very much for your valuable questions. As in "Syndicated bandits: A framework for auto-tuning hyper-parameters in contextual bandit algorithms", in our work we also assume that the arms in the arm set $X_t$ are drawn IID from some unknown distribution. Therefore, without an oblivious adversary choosing our arm set, we can anticipate that the expected performance of the bandit algorithm should be similar with close hyperparameters given other conditions (F_t) fixed. We will emphasize this point in our problem setting in the revision.
>
> **Weakness 3:**
>
> As we mention in our response to Weakness 1, in "Syndicated bandits: A framework for auto-tuning hyper-parameters in contextual bandit algorithms", the general regret bound is of order $\tilde{O}(T^{2/3} + \sqrt{MT})$. Their algorithm can only achieve the $\tilde{O}(\sqrt{T} + \sqrt{MT})$ and $\tilde{O}(T^{4/7}+ \sqrt{MT})$ bound of regret when there is **only one** hyperparameter to be tuned with some assumptions held. Specifically, to obtain the order $\tilde{O}(\sqrt{T} + \sqrt{MT})$, their work has to assume that the theoretical optimal exploration rate is smaller than any element in the candidate set. This assumption is stringent and unrealistic since it is widely known that the theoretical exploration rate is very large and conservative, and hence the elements in the candidate set should be smaller than the theoretical value in practice. On the other hand, we can observe that their regret bound also depends on the number of candidates $M$ in the order $\sqrt{MT}$, and $M$ becomes infinitely large under our problem setting. This fact implies that all their regret bounds are actually infinitely large and hence meaningless under our problem setting.
>
> We sincerely value the time you've dedicated to reviewing our work. And we respectfully hope you could re-evaluate our work if your concerns have been decently addressed.

---

> > ### Comment · Reviewer_psHy · 2023-11-20
> >
> > Thank you for the response.
> >
> > **Weak 1:**
> > Yes, the issue of handling a large base of algorithms is indeed challenging. However, this paper focuses primarily on contextual bandits, which typically involve a smaller number of parameters. For instance, in the mode selection problem within contextual bandits, we deal with no more than $\log d$ parameters.
> >
> > **Weak 2:**
> > The IID assumption may be applicable. However, in scenarios where $X_t$ is an IID sample from some unknown distribution, the optimal regret should be quantified as $\sqrt{Td\log K}$.
> >
> > **Weak 3:**
> > My reference was specifically to the $T^{2/3}$ regret results.

---

> ### Author Response · Authors · 2023-11-21
> **Thank you very much for your feedbacks**
>
> **Weakness 1:**
>
> Thank you very much for your valuable comments. Note $M$ in our rebuttal refers to the number of possible candidates for each hyperparameter. For example, $M$ could be the number of possible choices for the hyperparameter "exploration rate". To make it more clear, the general regret bound of “Syndicated bandits: A framework for auto-tuning hyper-parameters in contextual bandit algorithms” can be formulated as:
>
> $$\tilde O(T^{2/3} + \sum_{i=1}^L\sqrt{n_lT})$$
>
> where $L$ is the number of hyperparameters (usually small for contextual bandits) and $n_l  (\text{we call it }M)$ is the size of candidate value set for the $l$th hyperparameter. Here we consider tune each hyperparameter in an interval, which means the candidate set for each hyperparameter is a continuous interval and hence contains infinitely many candidates ($M = +\infty$). Note all the existing work, e.g. Syndicated, will incur linear regret under our problem setting since $M = +\infty$. In Appendix A.4.3, we firmly validate the necessity of tuning hyperparameters in a continuous set instead of a user-defined set with extensive experiments: finding the optimal hyperparameter combination for different bandit algorithms is a black-box problem and hence it is impossible for us to construct decent value candidate sets for all hyperparameters. If we discretize the interval finely, then the large size of the candidate set would evidently hurt the performance, as shown in the experiments. On the other hand, our proposed CDT could adaptively “zoom in” on the regions containing this optimal hyperparameter value automatically, without pre-specifying the candidate set.
>
> **Weakness 2:**
>
> Thank you for your further comments. Yes, we acknowledge that the optimal regret bound is of order $\tilde O(\sqrt{T})$. However, as in “Syndicated bandits: A framework for auto-tuning hyperparameters in contextual bandit algorithms”, it is inevitable to suffer from the suboptimal regret bound due to the enormous difficulty of bandit hyperparameter tuning in theory. Note “Syndicated bandits: A framework for auto-tuning hyperparameters in contextual bandit algorithms” also considers the generalized linear contextual bandits with theoretical guarantee. Furthermore, we consider a more challenging problem in our work with infinitely many candidate values for each hyperparameter, and under this problem setting all the existing theory in Syndicated and other general bandit model selection problems will be futile. Moreover, as we mention in Section 4.1, our work is more practical oriented and the extensive experiments in our work significantly validate the high efficiency of our algorithm over all baselines. And we include a non-trivial theoretical analysis for the completeness of our work in theory.
>
> **Weakness 3:**
>
> Thank you very much for your clarification. On the one hand, as we mentioned in our response to Weakness 1, the regret bound of Syndicated is $\tilde O(T^{2/3} + \sum_{i=1}^L\sqrt{n_lT})$, which becomes infinitely large under our much more difficult problem setting ($M = + \infty$). And we believe it is not very fair to compare our regret bound with the one in Syndicated since they considered an easier setting with finite $M$. On the other hand, even under the stochastic bandits, for continuum-armed bandits it is impossible to recover most of the results in multi-armed bandits in the same order. As the example in our first response to Weakness 1, the minimax regret bound for multi-armed bandit is of order $\sqrt{T}$ while the minimax regret bound for the continuum-armed bandit in $[0,1]^d$ is of order $T^{(d+1)/(d+2)}$. This fact also implies that it is intrinsically inevitable to get a worse regret bound when we consider the continuous set for hyperparameter tuning.
>
> We sincerely appreciate your valuable feedbacks on our work. Please let us know if our response has sufficiently resolved your concern and improved your opinion of our work, and we are more than happy to engage in any further discussion with you.

---

> ### Author Response · Authors · 2023-11-23
> **Thank you very much for your review. We are more than happy to receive feedbacks from you**
>
> Thank you very much for taking your time to review our work and raise insightful comments. In our last feedbacks, we have carefully examined your questions and comments, and then presented a solid response to your concerns.
>
> Specifically, we first clear up some potential misunderstandings from you regarding our work: in our problem setting, each hyperparameter can take any value in a continuous interval (infinitely many candidate values), and our algorithm can tune multiple hyperparameters simultaneously. In other words, $M$ in our response stands for the number of candidate values for each hyperparameter, instead of the number of hyperparameters we want to tune. Note all the regret bounds in existing literature (e.g. "Syndicated") will becomes infinitely large since their values depend on the number of candidate values $M$ for each hyperparameter. Not to mention that how to choose the candidate values is a black-box problem and hence unclear. Then, we also emphasize the significance of our theoretical analysis under a much more difficult setting, where all the existing theoretical results will fail.
>
>
> We sincerely appreciate your valuable feedbacks on our work. Since the discussion deadline is approaching, we respectfully hope you could let us know whether our response has sufficiently resolved your concerns and improved your opinion of our work. And we are more than happy to engage in any further discussions with you.

---

### Meta-Review · Area_Chair_A3hN · 2023-12-14

**Metareview:**

The paper presents approaches for effective hyper-parameter tuning for generalized linear contextual bandits in the stochastic setting. Empirical results are presented to support the work. While the reviewers mostly agree that the work makes progress on an interesting and practical problem, they share concerns ranging from potential computational issues due to the hierarchical nature of the approach to the weaker bounds compared to the basic setting, among others. The authors have provided responses to the specific questions, but the concerns linger on.

**Justification For Why Not Higher Score:**

There were certain concerns on the original submission, which does not seem to have been resolved to the satisfaction of the reviewers.

**Justification For Why Not Lower Score:**

n/a

---

### Decision · Program_Chairs · 2024-01-16

Reject